# A New Approach to Group Multi-Objective Optimization under Imperfect Information and Its Application to Project Portfolio Optimization

Eduardo Fernández [1], Nelson Rangel-Valdez [2,*], Laura Cruz-Reyes [3] and Claudia Gomez-Santillan [3,*]

1 Dirección de Investigación y Posgrado, Universidad Autonoma de Coahuila, Saltillo 26200, Mexico; eduardo.fernandez@uadec.edu.mx
2 División de Estudios de Posgrado e Investigación, Cátedras CONACyT—Tecnológico Nacional de México, Instituto Tecnológico de Ciudad Madero, Los Mangos 89440, Mexico
3 División de Estudios de Posgrado e Investigación, Tecnológico Nacional de México, Instituto Tecnológico de Ciudad Madero, Los Mangos 89440, Mexico; lauracruzreyes@itcm.edu.mx
* Correspondence: nelson.rangel@itcm.edu.mx (N.R.-V.); claudia.gomez@itcm.edu.mx (C.G.-S.)

**Abstract:** This paper addresses group multi-objective optimization under a new perspective. For each point in the feasible decision set, satisfaction or dissatisfaction from each group member is determined by a multi-criteria ordinal classification approach, based on comparing solutions with a limiting boundary between classes "unsatisfactory" and "satisfactory". The whole group satisfaction can be maximized, finding solutions as close as possible to the ideal consensus. The group moderator is in charge of making the final decision, finding the best compromise between the collective satisfaction and dissatisfaction. Imperfect information on values of objective functions, required and available resources, and decision model parameters are handled by using interval numbers. Two different kinds of multi-criteria decision models are considered: (i) an interval outranking approach and (ii) an interval weighted-sum value function. The proposal is more general than other approaches to group multi-objective optimization since (a) some (even all) objective values may be not the same for different DMs; (b) each group member may consider their own set of objective functions and constraints; (c) objective values may be imprecise or uncertain; (d) imperfect information on resources availability and requirements may be handled; (e) each group member may have their own perception about the availability of resources and the requirement of resources per activity. An important application of the new approach is collective multi-objective project portfolio optimization. This is illustrated by solving a real size group many-objective project portfolio optimization problem using evolutionary computation tools.

**Keywords:** group multi-objective optimization; multi-criteria classification; project portfolio selection; interval mathematics; evolutionary algorithms

## 1. Introduction

Frequently, real-life decision problems need several or many decision-makers (analysts, experts, stakeholders, participants, voters, etc.). A group decision-making problem (GDM) is described as a decision case in which a group of decision-makers (DMs) recognize the existence of a collective problem and are interested in finding a common action (alternative) that could be accepted as a good agreement.

Since the early Condorcet and Borda's works, including the seminal Arrow's impossibility theorem, GDM has attracted the attention of researchers for a very long time to the present date (e.g., [1–4]). Imprecision and vagueness are important issues in GDM. Since GDM is a process carried out by humans, fuzzy set-based methods, with their ability to model vagueness of human judgments and preferences, are especially suitable to address such problems. For a recent paper that revisits fuzzy and linguistic decision-making, the

reader is referred to [5]. Some other modern challenges faced by researchers in GDM are: (i) large-scale problems (LSGDM) involving more than twenty decision-makers (e.g., [6,7]); and (ii) manipulation and group dictatorship (e.g., [8]). These challenges have gained increasing importance as a consequence of the modern digital economy and the relevance of social networks.

The main processes for solving GDM are the consensus reaching process (CRP) and the selection process (SP) [9–11]. CRPs have the objective of obtaining the maximum degree of satisfaction from the DMs with the collective decision [9]. The search for consensus is an active process, with repetitive interactions (rounds, consensus steps) among group members, which contains a measure of consensus and a feedback method [10]; the last one suggests whether the group members should change their preferences, beliefs, and judgments to allow a better agreement [11,12]. Usually, the CRP is guided by a moderator [13]. The final solution is found through an SP, considering the preferences, beliefs, and judgments of the group members. Frequently, the SP consists of two phases: (A) aggregation of preferences, beliefs, and judgments from the members; and (B) use of these preferences, beliefs, and judgments, that were aggregated collectively, to find a solution which should correspond, as much as possible, to the aggregated group opinions [9,14]. In this paper, our interest is limited to problems in which the decision alternatives are described by multiple criteria, the so-called multi-person-multi-criteria decision making [5]. Here, the aggregation of the group member's preferences on conflicting criteria plays a crucial role in identifying an acceptable collective agreement.

Usually, the selection process concerns a relatively small set of options; many methods have been proposed to address such cases (e.g., [3,4,15–17]). Alternatively, the group multi-objective decision problem on very large decision sets (characterized by constraints) has received comparatively very little attention. According to [18], we call this problem GDM-MOP, denoting the combination of group decision making and multi-objective optimization.

To the best of our knowledge, the existing GDM-MOP approaches are not free of some of the following criticisms:

1. Most interactive methods implicitly assume that group preferences are transitive and comparable relations, although the lack of transitivity is a well-established characteristic of voting systems (e.g., [19]). Even in the case of a single DM, transitivity, and comparability of their preference relation are subject to question, mainly in the presence of veto conditions, and/or when the number of objectives overcomes the cognitive limitations of the human mind.

2. Many methods are susceptible to manipulation. According to classical voting theory, under very general conditions, every voting procedure can be manipulated by some voters by declaring insincere preferences (e.g., [19]).

3. Popular interactive approaches help to obtain acceptable agreements because each DM learns the preferences from the other DMs and correspondingly fits their own. However, the final accepted solution may significantly differ from those that each DM would have considered as satisfactory if the decision had depended solely of them. Thus, the consensus does not result from the search in the set of possible solutions but from mutual concessions. Group satisfaction is partial because it is only achieved by recognizing that a more satisfactory result is not possible.

4. The handle of imprecision, uncertainty, and ill-definition in GDM-MOP is a real concern. GDM-MOP approaches typically assume that the whole group agrees on the resource availability, the resource consumption, and objective values for each point in the decision variable space. However, there could be several sources of imperfect information which affect that assumption. Indeed, each DM may have their own perception (no free of certain imprecision, uncertainty, or arbitrariness) about objective values, available and required resources. Such imperfect knowledge may impact the individual best solutions, on the collective preferences, and the consensus degree. Under imperfect information, the consensus search process is even more

difficult and relevant since the diverse perceptions from the DMs and different levels of conservatism should be aggregated and, if possible, agreed.

5.   In complex problems, some DMs with very different value systems and/or roles with respect to the other group members may consider different sets of objective functions and constraints. Such a case is not addressed by most of the methods to solve GDM-MOPs.

This paper presents a proposal that strongly reduces the above criticisms. Since group (dis)satisfaction depends on the number of its (dis)satisfied members, the paper focuses on a consensus measure based on counting the number of DMs who are satisfied (respectively dissatisfied) with their respective objective values on a common point in the decision variable space. For each point in the feasible decision set, satisfaction or dissatisfaction from each group member is determined by a multi-criteria ordinal classification approach, based on comparing solutions with a limiting boundary between classes "unsatisfactory" and "satisfactory". By counting the group members who are (dis)satisfied with solutions in the decision variable space, group satisfaction and dissatisfaction can be simultaneously optimized. This paper has perhaps the merit to be the first in addressing consensus evaluation as multi-criteria ordinal classification problems solved by the entire set of group members. Consensus is identified by a high level of group satisfaction and a low level of dissatisfaction. The group moderator is in charge of making the final decision, finding the best compromise between collective satisfaction and dissatisfaction.

Interval numbers are used to model imperfect information from each group member. This model is less sophisticated than more general fuzzy approaches, but, to a great extent, without their mathematical complexity. Choosing interval numbers as the model of imprecisions allows using recently proposed interval-based multi-criteria ordinal classification methods. Two different decision models are proposed in this paper; the first one uses the interval outranking approach and INTERCLASS-nB method by Fernandez et al. in [20,21]; the second one is based on building and exploiting an interval weighted-sum value function. On one hand, the outranking model is appropriate for handling non-compensatory preferences, allowing veto effects and incomparability situations. On the other hand, the weighted-sum function is, to a certain extent, a simpler and rougher model, which may be recommended to represent compensatory preferences. With a little loss of generality, we address multi-objective maximization problems under resource constraints (as typically in project portfolio optimization), but the method can be trivially extended to other kinds of problems.

The structure of the paper follows: In Section 2, we outline the background on which our proposal is based. Section 2.1 refers to several relevant precedent papers. The concept of good consensus, some fundamental aspects about interval numbers, and an interval-based multi-criteria classification method are briefly described in Section 2.2, Section 2.3, Section 2.4. The problem is detailed in Section 3. Sections 4 and 5, combined with the concept of maximum consensus, form the core of the proposal. These sections present the mathematical model of what a satisfied/dissatisfied DM is from two points of view: the outranking model of preferences (Section 4) and the weighted-sum model (Section 5). The method is summarized in Section 6; the whole proposal is illustrated by a real-size many-objective project portfolio optimization problem in Section 7. Lastly, several concluding remarks are discussed in Section 8. Two evolutionary algorithms used in solving optimization problems are described in appendices.

## 2. Background

### 2.1. An Overview of GDM-MOP Literature

In the GDM-MOP field, some approaches obtain a representative Pareto sample and then apply a method to aggregate individual preferences in a model of collective preference, which is used to find a final solution (e.g., [22,23]). Other popular approaches propose an integration of group preference handling with interactive procedures of multi-objective optimization (e.g., [24–26]). In some methods, the interaction is performed during the

optimization process. In other methods, the interaction is performed once a Pareto sample has been generated (e.g., [27]).

Efremov et al. in [28] developed a decision support system for e-democracy inspired on Pareto frontier visualization, goal identification, and arbitration. Collective intelligence methods to aggregate reference points from different DMs have been proposed in ([29–32]) to approach the region of the Pareto frontier that is more preferred by the DMs. Bechickh et al. (2013) proposed a negotiation support system that includes the DMs' preferences through reference points; priorities on the set of DMs are characterized by weights; the output of a negotiation round is a single group reference point. Xiong et al. (2013) introduced fuzzy reference points; this approach contributes to improving the robustness of the final solution dealing with imprecise and changing preferences.

Fernandez and Olmedo in [33] search for good consensus solutions by maximizing, (respectively, minimizing), the number of group members who are satisfied (resp. dissatisfied) with the current solution in the decision variable space. To solve the bi-objective optimization problem, they used the NSGA2 evolutionary algorithm. The DMs are declared as (dis)satisfied by comparing the current solution with their best compromise.

The NEMO-GROUP, a set of interactive evolutionary multi-objective optimization (MOO) methods, was developed by Kadzinski and Tomczyk in [34]. In these approaches, an evolutionary algorithm is modified with the introduction of pairwise comparisons of several DMs. Solutions are evaluated using utilitarian and egalitarian additive group value functions; the evolutionary algorithms accept weights assigned to the DMs.

Borissova and Mustakerov in [35] presented a two-step placement algorithm, which combines MOO and GDM. First, MOO is used to identify a set of design alternatives for object placement. In the second step, business intelligence and group decision making are used to evaluate design alternatives.

A particular application of MOO is the distributed engineering design, the complex systems built in this area are getting more popularity (e.g., [36]). These systems are often designed by a group of DMs; the design of each subsystem is in charge of a DM. The design teams only have partial information about the global design, and negotiations are relevant to reach a consensus. Guarneri and Wiecek in [37] developed a mathematical model of the problem considering the aspect of distribution and decomposition. The solution of the model is based on Lagrangian relaxation.

An important case of GDM-MOPs is group multi-objective project portfolio optimization. Project portfolio selection is one of the most important problems faced by top-level managers in large enterprises and public organizations. Often, projects are described by several (or many) conflicting criteria, and the selection of the "best" portfolio should be made by a collective entity. This group may be composed of experts in different/complementary fields or members of the top level management. The group members share an interest in finding the portfolio that could bring the best results for the organization, but they have different value systems, beliefs, preferences on conflicting criteria, and judgments about resource consumption and availability. Individual preferences and beliefs are typically modeled by certain parameters and values in a decision model; they can significantly differ from different DMs. If the whole group is seen as an entity, the dispersion of model parameters and values can be considered as imperfect information in the sense of Roy et al. (2014). According to such a paper, imperfect knowledge comes from arbitrariness, imprecision, ill-determination, and uncertainty in data and model parameters ([38]). In project portfolio optimization, imperfect information concerning weights and criterion scores has been addressed by Liesio et al. in [39,40], Fliedner and Liesio in [41], Toppila and Salo in [42], and Balderas et al. in [43] using interval mathematics, although no one of these papers concern the specific characteristics of GDM.

### 2.2. Toward a Maximum Consensus

According to Fernández and Olmedo (2013), in GDM, reaching an acceptable agreement is not always possible. This comes from strong contradictions amongst important

disjoint subsets of group members; for instance, there could be a clear opposition of a numerically significant minority that does not want to accept a preference coming from a not very strong majority of the group. Reaching a collective agreement is only possible when there is unanimity or an appreciable level of consensus. Among many ways to address and define consensus (e.g., [10,14,44–47]), Fernandez and Olmedo (2013) consider a good consensus as associated with the fulfillment of two conditions:

(A)　There is an important agreeing majority with a particular alternative (or solution);
(B)　There is no appreciable disagreeing minority.

With Condition (B), Fernandez and Olmedo in [33] and Fernandez et al. in [48] rejected "majority dictatorship", which neglects the importance of the intensity of disagreement and degrades fairness and equity concerning minorities.

Let us consider a group with $n_g$ members under the following general premise:

Fundamental Premise: Each solution $z$ in the decision variable space (with its corresponding objective values and feasibility) can be classified by the $i$-th DM into one and only one of the following classes: (i) $z$ is a satisfactory solution; (ii) $z$ is an unsatisfactory solution; (iii) the $i$-th DM is neither satisfied nor dissatisfied with $z$.

Point (iii) concerns situations in which a DM hesitates about the appropriate classification of a solution and confers to the above premise a very high generality.

Let $N_{sat}$ (resp. $N_{dis}$) denote the number of group members who are satisfied (resp. dissatisfied) by a particular solution $z$. $N_{sat}$ and $N_{dis}$ are integer functions of $z$. The ideal consensus (if possible) should correspond to a point $z^*$ such that $N_{sat}(z^*) = n_g$ and $N_{dis}(z^*) = 0$.

Therefore, the ideal consensus is the ideal solution to the bi-objective optimization problem:

$$\text{Maximize } N_{sat}(z), \text{ Minimize } N_{dis}(z) \tag{1}$$

Unlike typical multi-objective optimization problems in which its ideal solution is not feasible, Problem 1 may reach the ideal point $N_{sat}(z^*) = n_g$ and $N_{dis}(z^*) = 0$ as much as the preferences and beliefs of all the individual DMs become close enough through an effective consensus reaching process. Even when the ideal consensus is not possible, good agreements can be identified when $N_{sat} \geq 2/3\ n_g$ (Condition A) and $N_{dis} \approx 0$ or $N_{dis} << 0.5\ n_g$ (Condition B).

For a given closeness of preferences and beliefs of the individual DMs, the best consensus is a non-dominated solution to Problem 1. If this current consensus level is not high enough, a new CRP step should allow identifying better agreements.

To make operational the process of solving Problem 1, we need a mapping between the decision variable space and the values $N_{sat}$ and $N_{dis}$. Therefore, we require a model that permits, for each DM and each $z$ in the decision variable space, knowing to which class of satisfaction/dissatisfaction $z$ is assigned to. Most of the remaining paper is devoted to discussing some preference models and assignment procedures to this end. Since in the objective space, $z$ is described by multiple objective values, its assignment should be considered as a multi-criteria ordinal classification problem.

### 2.3. Some Fundamental Notions on Interval Mathematics

Moore in [49] defines an interval number as an extension of the concept of a real number and as a subset of the real line $\mathbb{R}$, describing it as a range, $\boldsymbol{E} = [\underline{E} + \overline{E}]$, where $\underline{E}$ is the lower limit and $\overline{E}$ is the upper limit. In the rest of this paper, an interval number is denoted with boldface italic letters.

Let $\boldsymbol{D}$ and $\boldsymbol{E}$ two interval numbers and $F$ a real number. Basic arithmetic operations are defined below.

$$\boldsymbol{D} + \boldsymbol{E} = \left[\underline{D} + \underline{E}, \overline{D} + \overline{E}\right] \tag{2}$$

$$\boldsymbol{D} - \boldsymbol{E} = \left[\underline{D} - \overline{E}, \overline{D} - \underline{E}\right] \tag{3}$$

$$\boldsymbol{D} \cdot \boldsymbol{E} = \left[min\{\underline{D} \cdot \underline{E}, \underline{D} \cdot \overline{E}, \overline{D} \cdot \underline{E}, \overline{D} \cdot \overline{E}\},\ max\{\underline{D} \cdot \underline{E}, \underline{D} \cdot \overline{E}, \overline{D} \cdot \underline{E}, \overline{D} \cdot \overline{E}\}\right] \tag{4}$$

$$\boldsymbol{D} \cdot F = \left[\underline{D} \cdot F, \overline{D} \cdot F\right] \tag{5}$$

A possibility measure of an order relation over interval numbers is given by Equation (6) [50].

$$
P(\boldsymbol{E} \geq \boldsymbol{D}) = \left\{ \begin{array}{ll} 1 & \text{if } p_{ED} > 1, \\ p_{ED} & \text{if } 0 \leq p_{ED} \leq 1, \\ 0 & \text{if } p_{ED} \leq 0 \end{array} \right.
\tag{6}
$$

$$
\text{where } p_{ED} = \frac{\overline{E} - \underline{D}}{(\overline{E} - \underline{E}) + (\overline{D} - \underline{D})}
$$

$$
\text{If } \overline{e} = \underline{e} = e \text{ and } \overline{d} = \underline{d} = d \; P(\boldsymbol{E} \geq \boldsymbol{D}) = \left\{ \begin{array}{l} 1 \text{ if } e \geq d, \\ 0 \text{ otherwise.} \end{array} \right.
\tag{7}
$$

According to Fliedner and Liesio (2016), a real number $e$ within the interval $[\underline{E}, \overline{E}]$ is said to be a realization of the interval number $\boldsymbol{E}$. In [20], P $(\boldsymbol{D} \leq \boldsymbol{E}) = \alpha$ is interpreted as the degree of credibility that once two realizations are given from $\boldsymbol{E}$ and $\boldsymbol{D}$, the realization $d$ will be smaller than or equal to the realization $e$.

According to Fernández et al. (2019), the possibility function satisfies Equation (8). This equation is combined with the interpretation of the order relation in Equation (9).

$$
P(\boldsymbol{E} \geq \boldsymbol{D}) = \alpha \Rightarrow P(\boldsymbol{D} \geq \boldsymbol{E}) = 1 - \alpha \text{ (negation)}
\tag{8}
$$

$$
\boldsymbol{E} > \boldsymbol{D} \Leftrightarrow P(\boldsymbol{E} \geq \boldsymbol{D}) > 0.5 \text{ (strict order)}
\tag{9}
$$

A more reliable strict order (denoted by $>_\alpha$) on interval numbers is defined in Equation (10).

$$
\boldsymbol{E} >_\alpha \boldsymbol{D} \iff P(\boldsymbol{E} \geq \boldsymbol{D}) \geq \alpha > 0.5
\tag{10}
$$

### 2.4. Multi-Criteria Ordinal Classification Based on an Interval Outranking Approach

There are more than one hundred multi-criteria decision-making (MCDM) methods. According to the model of the decision-maker's preferences, most of the methods can be grouped in one of three basic paradigms:

- The functional paradigm, based on building functions that model the decision-maker's preferences (e.g., [51,52]);
- The relational paradigm, based on building crisp or fuzzy binary relations (e.g., [53]);
- The symbolic paradigm, which is mainly related to artificial intelligence (e.g., [54]).

Whatever the model of the DM's preferences, we can distinguish three MCDM fundamental problems: multi-criteria choosing, ranking, and sorting. Each MCDM method addresses one of these general problems. For choosing and ranking, multi-attribute utility theory (MAUT) ([51]) and the analytic hierarchy process (AHP) ([52]) are the most popular approaches. Although Ishizaka et al. in [55] proposed a variant of AHP for multi-criteria sorting (also called multi-criteria ordinal classification) problems, outranking and symbolic methods are more popular for this purpose.

Perhaps the most known multi-criteria ordinal classification method is ELECTRE TRI-B ([56]), which was generalized to ELECTRE TRI-nB in ([57]). Such methods make use of an outranking relation, which means that for a pair of decision actions $(x,y)$, we should check the validity of the assertion "action (alternative) $x$ is at least as good as action (alternative) $y$". Two effects (concordance and discordance) are combined to determine the degree of credibility of the outranking (cf. [53]). In ELECTRE TRI-B and nB, the boundary between adjacent classes is described by limiting actions. Unlike ELECTRE TRI-B, its extension allows characterizing the boundary between adjacent classes by several limiting profiles. Recently, ELECTRE TRI-nB was extended to handle imperfect information in terms of interval numbers ([21]). The so-called INTERCLASS-nB shares similar assignment rules and consistency properties with ELECTRE TRI-nB, but using the interval outranking approach proposed in ([20]). This handles imperfect information on criterion scores, weights, veto thresholds, and majority thresholds with interval numbers. Even missing criterion scores can be handled by the approach. According to Fernández et al. (2020), "eliciting model

preference parameters generally suffer from arbitrariness and imperfect knowledge (in particular, due to ill-determination). This occurs in a more acute way when there is a group of DMs, so that conflicting views are quite natural and frequent."

The interval outranking approach and INTERCLASS-nB work with several interval preference relations:

**Definition 1.**

*(i)　　$xS(\beta,\lambda)y \Leftrightarrow \sigma(x,y,\lambda) \geq \beta$ (interval outranking);*
*(ii)　　$xP_r(\beta,\lambda)y \Leftrightarrow \sigma(x,y,\lambda) \geq \beta$ and $\sigma(y,x,\lambda) < \beta$ (interval preference);*
*(iii)　$xI(\beta,\lambda)y \Leftrightarrow \sigma(x,y,\lambda) \geq \beta$ and $\sigma(y,x,\lambda) \geq \beta$ (interval indifference);*

*where:*

*$\sigma$ is the credibility index of the interval outranking;*
*$\lambda$ is an interval number representing a majority threshold; $\lambda > [0.5, 0.5]$ and $\lambda^{min} \geq 0.5$;*
*$\beta$ is a credibility threshold for establishing a credible crisp outranking relation; $\beta > 0.5$.*

The INTERCLASS-nB, adapted to the case of only two ordered classes, is presented below:
*Conditions on the limiting profiles: Let $C_1$ and $C_2$ be ordered classes ($C_2$ is the most preferred). Set $\beta > 0.5$ and $\lambda > [0.5, 0.5]$ with $\lambda^{min} \geq 0.5$. The boundary between $C_1$ and $C_2$ is characterized by a set of limiting profiles, $B = \{b_j\}$, such that:*

*(i)　　All $b_j$ of $B$ belongs to $C_2$;*
*(ii)　There is no pair $(b_j, b_i)$ such that $b_j P_r(\beta,\lambda)b_i$.*

The method requires the definition of interval outranking and preference relations between actions and the limiting boundary, as follows:

**Definition 2.**

*(i)　　$xS(\beta,\lambda)B$ iff there is a $w \in B$ such that $xS(\beta,\lambda)w$ and there is no $y \in B$ with $yP_r(\beta,\lambda)x$;*
*(ii)　$BP_r(\beta,\lambda)x$ iff there is a $w \in B$ such that $wP_r(\beta,\lambda)x$ and there is no $y \in B$ with $xP_r(\beta,\lambda)y$.*

The "pessimistic" and "optimistic" assignment rules are:

**Definition 3 ("pessimistic" rule).** *Suppose Set B satisfies the conditions on the limiting profiles.*

*Step 1: Compare x to B;*
*Step 2: If $xS(\beta,\lambda)B$, then assign x to class $C_2$;*
*Step 3: If $not(xS(\beta,\lambda)B)$, then assign x to $C_1$.*

**Definition 4 ("optimistic" rule).** *Suppose Set B satisfies the conditions on the limiting profiles.*

*Step 1: Compare x to B;*
*Step 2: If $BP_r(\beta,\lambda)x$, then assign x to class $C_1$;*
*Step 3: If $not(BP_r(\beta,\lambda)x)$, then assign x to $C_2$.*

It is easy to prove that the "optimistic" rule always suggests an assignment to a class not worse than the one suggested by the "pessimistic" procedure.

For a detailed description of INTERCLASS-nB, the reader is referred to [21].

## 3. Characterization of GDM-MOPs Under-Study

In this paper, we address group multi-objective optimization problems described by the following characteristics:

a.　There is a group moderator who is in charge to control and guide the consensus reaching process;
b.　The individual DMs participate in the decision process providing information about their preferences, beliefs, and level of conservatism, and modifying this information during consecutive steps of the CRP;
c.　Some (even all) objective values may be not the same for different DMs;

d.   Each DM may handle a different set of objective functions;
e.   The objective values may be imperfectly known (subject to imprecision or uncertainty);
f.   The availability of resources may be imperfectly known;
g.   Resource requirements per activity (project, in case of portfolio optimization) may be imperfectly known;
h.   Each group member has their own opinion about the availability of resources and requirements per activity (project, in case of portfolio optimization);
i.   All the DMs consider a common point in the decision variable space.

Points c, d, e, f, g, and h imply that each group member faces their own MOO problem under imperfect knowledge. The model of such imperfect knowledge is based on the following assumption:

**Assumption 1.** *Each DM is willing to represent as interval numbers the imperfect information on objective values, availability, and requirement of resources.*

In the framework of project portfolio optimization, the above statement has been assumed by Liesio et al. (2007, 2008), Fledner and Liesio (2016), Toppila and Salo (2017), and Balderas et al. (2019). This assumption is a bit more restrictive than it appears. As stated by [41], if we assume that a magnitude is an interval number, one should admit that (i) this magnitude can take any value within this interval, and (ii) the DM does not privilege the credibility of any particular value within the range.

Let $AR^j_i$ be the interval number used to represent the estimated availability of the $j$-th resource from the point of view of the $i$-th DM; $NR^j_i$ is the interval number used to denote the aggregation of required resources, which depends on the decision variables vector $z$.

In order to find a feasibility condition for the $i$-th DM, a solution $z$ must satisfy $AR^j_i > \xi^i NR^j_i(z)$ according to the order relation given by Equation (10). The $i$-th DM faces the MOP in Equation (11).

$$\underset{z \in R_{F_i}}{\text{Maximize}} F_i = (f_{1i}(z), f_{2i}(z), \ldots, f_{Ni}(z)) \tag{11}$$

where $N_i$ is the number of the evaluation criterion for the $i$-th member of a group; $F_i$ is the vector function being considered by the $i$-th DM; $z$ is the vector of decision variables; in project portfolio optimization, $z$ is a vector composed of binary variables, in which "1" means "the related project belongs to the portfolio", and "0" otherwise.

$R_{Fi}$ is the feasible region from the point of view of the $i$-th DM. It is defined by the interval-based resource constraints $AR^j_i > \xi^i NR^j_i(z)$, and perhaps other non-resource constraints.

In Problem 11, $\xi^i$ is the credibility threshold for the expression "the available resources are sufficient to satisfy the requirements". The more conservative the $i$-th DM, the higher the value of $\xi^i$.

**Definition 5 ($\alpha$-dominance).** *Let $O_i$ be the image of the feasible region of Problem 11. Consider an element (x,y) belonging to $O_i \times O_i$, where $x \neq y$, and $\alpha$ a real number in [0.5, 1]. The solution y is $\alpha$-dominated by x (denoted by $xD(\alpha)y$) iff $\min\{P(f_{ji}(x) \geq f_{ji}(y)), j = 1, \ldots, N_i\} = \alpha$ and $P(f_{ki}(x) \geq f_{ki}(y)) > 0.5$ for some k.*

Identifying satisfactory solutions is a preference judgment. Such a judgment is related to what the DMs consider as their maximum aspiration levels, which come from the best solution to Problem 11. Therefore, the DM should be able to determine such levels.

The best solution to Problem 11 is ill-defined due to the conflicting nature of its objective functions. The following two definitions are useful to characterize a necessary condition to be the best compromise solution.

**Definition 6 (a preferred solution by the i-th DM).** *A solution $x \in O_i$ is preferred to $y \in O_i$ by the i-th DM if the following statement is considered as true by them: "x is at least as good as y, but y is not at least as good as x".*

**Remark 1.** *The interval preference relation from Definition 1 ($xP_r(\beta,\lambda)y$) and the dominance relation from Definition 5 ($xD(\alpha)y$) are good arguments in support to "x is preferred to y" for sufficiently high values of $\beta$ and $\alpha$.*

**Definition 7 (necessary conditions to be the best compromise solution for the i-th DM).** $x_i^* \in O_i$ *is the best compromise solution to Problem 11 for the i-th DM only if it fulfills two conditions:*

(A)    *There is no y in $O_i$ such that y is preferred to $x_i^*$ by the i-th DM;*
(B)    *There are arguments to justify that the i-th DM considers $x_i^*$ as at least as good as many solutions that satisfy Condition A.*

**Assumption 2 (capacity to identify the best compromise solution from the i-th DM's point of view).** *Let $O_i$ be the image of the feasible region of Problem 11. Each individual DM is able to solve Problem 11, thus identifying their best compromise solution $x_i^* \in O_i$.*

The way in which the *i*-th DM can find their best compromise depends on the model of multi-criteria preferences; this should correspond to how the DM aggregates preferences on conflicting criteria. As stated in Section 2, there are three main paradigms: (i) the functional approach; (ii) the relational approach; and (iii) symbolic methods from Artificial Intelligence, mainly those based on rough sets theory. To the best of our knowledge, no symbolic multi-criteria decision method in the frame of interval numbers has been reported. The functional approach bases its prescriptions on a value or utility function, fulfilling axioms of full comparability and transitivity; these value or utility functions are compensatory models in which poor scores of some criteria can be compensated by good scores of other ones. There are a few papers that have used interval-based weighted sum functions as decision tools in project portfolio selection problems (e.g., [39–41]). To our knowledge, more complex forms of interval-based value functions have not been proposed to address multi-objective optimization problems; further, interval value functions have not been used in solving multi-criteria classification problems. The relational paradigm is generally focused on building and exploiting outranking relations; this approach can handle non-compensatory effects, veto situations, incomparability, and non-transitive preferences. In the frame of interval numbers, the method in ([20]) allows building credible interval outranking relations, and the INTERCLASS-nB method in ([21]) is able to suggest appropriate assignments in multi-criteria ordinal classification problem under imperfect information.

## 4. Model of Preferences and Judgments of a DM with a Non-Compensatory Aggregation of Preferences

In this section, we suppose that the *i*-th DM is compatible with the interval outranking model by Fernández et al. (2019). We refer to a generic DM, and for simplicity, we avoid the use of the subscript "*i*". Firstly, let us present an overview of the interval outranking approach from [20].

### 4.1. The Interval Outranking Model

The method requires the following assumption:

**Assumption 3 (modeling arbitrariness and ill-definition of parameters).** *The DM is willing to specify an outranking model in terms of weights, veto thresholds, a majority threshold, and a credibility threshold. In order to model imperfect information, those parameters can be specified as interval numbers. Therefore, the weights will be denoted as $w_j = [w_j^-, w_j^+]$, the veto thresholds as $v_j = [v_j^-, v_j^+]$, the majority threshold as $\lambda = [\lambda^-, \lambda^+]$, and the credibility threshold as $\beta = [\beta^-, \beta^+]$.*

The credibility index $\delta_j(a', a)$ of the assertion "$a'$ is at least as good as $a$ regarding criterion $f_j$" is

$$\delta_j(a', a) = P(f_j(a') \geq f_j(a)) \, j = 1, \ldots, N \tag{12}$$

where $P$ is the possibility function of Equation (6).

We say that criterion $f_j$ is $\gamma$-concordant with the statement "$a'$ is at least as good as $a$ regarding criterion $f_j$" (denoted $a'S_ja$) if and only if $a'S_ja$ with a credibility index of at least $\gamma$. The set of criteria $f_j$ such that $\delta_j(a', a) \geq \gamma$ ($\gamma = \min\{\delta_j\}$) is called $\gamma$-possible concordance coalition with "$a'$ is at least as good as $a$" and is denoted by $C(a'S_\gamma a)$. $\gamma$ is the credibility that all criteria in $C(a'S_\gamma a)$ agree that "$a'$ outranks $a$". All criteria not in $C(a'S_\gamma a)$ form the $\gamma$-possible discordance coalition, $D(a'S_\gamma a)$.

For consistency reasons, the interval weights should satisfy the following constraints:

$$\sum_{j=1}^{N} w_j^- \leq 1,$$
$$\sum_{j=1}^{N} w_j^+ \geq 1.$$

The concordance index of "$a'$ outranks $a$" is an interval number denoted by $c(a', a) = [c^-(a', a), c^+(a', a)]$, and calculated as follows:

$$c^-(a', a) = \sum_{fj \in C(a'S_\gamma a)} w_j^- \tag{13}$$

when

$$\sum_{fj \in C(a'S_\gamma a)} w_j^- + \sum_{fj \in D(a'S_\gamma a)} w_j^+ \geq 1$$

otherwise,

$$c^-(a', a) = 1 - \sum_{fj \in D(a'S_\gamma a)} w_j^+ \tag{14}$$

similarly,

$$c^+(a', a) = \sum_{fj \in C(a'S_\gamma a)} w_j^+ \tag{15}$$

when

$$\sum_{fj \in C(a'S_\gamma a)} w_j^+ + \sum_{fj \in D(a'S_\gamma a)} w_j^- \leq 1$$

otherwise,

$$c^+(a' \, a) = 1 - \sum_{fj \in D(a'S_\gamma a)} w_j^- \tag{16}$$

Since the set of criteria in $C(a'S_\gamma a)$ is determined by $\gamma$, the concordance index $c$ depends on such a value. Such a dependence is denoted by $c(a', a, \gamma)$.

Each possible concordance coalition $C(a'S_\gamma a)$ provides reasons in favor of "$a'$ outranks $a$", and each possible discordance coalition $D(a'S_\gamma a)$ provides reasons against it. In the following, we denote as $d_j(a',a)$ the credibility index of the statement "$a'Sa$ is vetoed by $g_j$". $d_j(a',a)$ is defined as $P(f_j(a) \geq f_j(a') + v_j)$, where $v_j$ is the interval veto threshold related to $f_j$.

The credibility of a possible outranking given a $\gamma$-possible concordance coalition and the corresponding discordance coalition is defined as follows:

**Definition 8.** *Let $\Omega$ be the set $\{\delta_j > 0, j = 1, \ldots, N\}$. Given $\gamma \in \Omega$ we say that $a'$ outranks $a$ for the $\gamma$-possible concordance coalition with credibility index $\sigma_\gamma$ and majority threshold $\lambda > 0.5$ with $\lambda^- > 0.5$ iff $\sigma_\gamma = \min\{\gamma, P(c(a', a, \gamma) \geq \lambda), (1 - \max_{fj \in D(a'S_\gamma a)} d_j(a',a))\}$.*

For each particular value of $\gamma$, this index can be interpreted as a credibility degree of the conjunction of two predicates: (i) "the $\gamma$-concordance coalition is strong enough" and (ii) "the $\gamma$-discordance coalition does not exert veto".

The above marginal (single coalition) credibility indices are merged by using the "*max*" operator in the following definition:

**Definition 9 (interval outranking credibility index).** *From the point of view of the interval outranking approach and under the above notation, a′ outranks a ("a′ is at least as good as a") with credibility index* $\sigma(a',a,\lambda) = max\{\sigma_\gamma\}$ *($\gamma \in \Omega$) and majority threshold $\lambda > 0.5$ with $\lambda^- > 0.5$. If $\Omega$ is empty, then $\sigma(a',a,\lambda)$ is zero.*

Let $\boldsymbol{\beta}$ an interval number ($0.5 < \boldsymbol{\beta} < 1$ with $\beta^- > 0.5$) considered as a credibility threshold to establish crisp preference relations. In the following we will use $a'S(\boldsymbol{\beta},\boldsymbol{\lambda})a$, $a'I(\boldsymbol{\beta},\boldsymbol{\lambda})a$, and $a'P_r(\boldsymbol{\beta},\boldsymbol{\lambda})a$ similarly to Definition 1, but using the interval number $\boldsymbol{\beta}$ instead of the real number $\beta$. Such a change gives more flexibility to the DM in setting the credibility threshold.

**Remark 2.**

(a)　As was proved in [20], if $f_j(a)$ are real numbers for $j = 1, \ldots , N$, then $aS(1,1)a$.
(b)　$a's(\boldsymbol{\beta},\boldsymbol{\lambda})a \Rightarrow a'S(\xi,\boldsymbol{\lambda})a \ \forall \ \xi < \boldsymbol{\beta}$.

**Proposition 1.** *Some properties of the interval dominance combined with the interval outranking and preference relations:*
　　*For all $(x,y,z) \in O \times O \times O$:*

i.　　$yD(\alpha)x$ and $xS(\boldsymbol{\beta},\boldsymbol{\lambda})z \Rightarrow yS(\varepsilon,\boldsymbol{\lambda})z$ for some $\varepsilon \geq min(\alpha,\boldsymbol{\beta})$
ii.　　$zP_r(\boldsymbol{\beta},\boldsymbol{\lambda})y$ and $yD(\alpha)x \Rightarrow zP_r(\varepsilon,\boldsymbol{\lambda})x$ for some $\varepsilon \geq min(\alpha,\boldsymbol{\beta})$
iii.　$yD(\alpha)x$ and $xP_r(\boldsymbol{\beta},\boldsymbol{\lambda})z \Rightarrow yP_r(\varepsilon,\boldsymbol{\lambda})z$ for some $\varepsilon \geq min(\alpha,\boldsymbol{\beta})$
iv.　$If \ \alpha > 0.5$, then $xD(\alpha)y \Rightarrow xP_r(\boldsymbol{\beta},\boldsymbol{\lambda})y$ for some $\boldsymbol{\beta} \geq \alpha$ and for all $\boldsymbol{\lambda} \leq [1,1]$
v.　　$xD(\alpha)y$ and $yD(\eta)z \Rightarrow xD(\varepsilon)z$ with $\varepsilon = min \ (\alpha,\eta)$

The proofs are similar to the ones in [20]. The single difference consists of replacing the real value of $\beta$ with the middle points of the interval number $\boldsymbol{\beta}$.

*4.2. Finding the Best Compromise solution to Problem 11*

According to Assumption 2, each DM is capable of finding their best solution to Problem 11. Here we discuss a method for this purpose.

**Definition 10 ((β,λ) non-strictly outranked solution).** *A solution $x \in O$ such that there is no $y \in O$ fulfilling $yP_r(\boldsymbol{\beta},\boldsymbol{\lambda})x$ is called a $(\boldsymbol{\beta},\boldsymbol{\lambda})$ non-strictly outranked solution. The $(\boldsymbol{\beta},\boldsymbol{\lambda})$ non-strictly outranked frontier of Problem 11 is the set of these solutions.*

**Remark 3.**

(i)　With appropriate values of $\boldsymbol{\beta}$ and $\boldsymbol{\lambda}$, a non-strictly outranked solution fulfills Condition A of Definition 7, that is, the first necessary condition to be the best compromise.
(ii)　Condition B of Definition 7 is proved on the non-strictly outranked frontier, using the outranking strength measure. This measure is described as $OS(x) = card \{a_i \in NSF$ such that $xS(\boldsymbol{\beta},\boldsymbol{\lambda})a_i\}$, where NSF denotes the $(\boldsymbol{\beta},\boldsymbol{\lambda})$ non-strictly outranked frontier.
(iii)　More than one solution can fulfill the necessary conditions of Definition 7. The solution selected as the final best compromise should be one with the highest value of the outranking strength.

Problem 11 can be solved through the following steps:

1. The individual DM (perhaps helped by a decision analyst) sets their model parameters according to Assumption 3.
2. The $(\beta, \lambda)$ non-strictly outranked frontier is identified by an optimization algorithm; the set of solutions that fulfill Definition 10 is determined;
3. The DM selects the best compromise solution according to Remark 3.iii.

The optimization algorithm for the step 2 depends on the problem characteristics. Balderas et al. (2019) proposed an evolutionary algorithm called I-NOSGA for solving project portfolio optimization problems with several or many interval-valued objective functions and with an outranking relation defined on the objective space. I-NOSGA separates each population into non-strictly outranked fronts. An extension of I-NOSGA to address the optimization problems in this paper is provided in Appendix A.

*4.3. Making Judgments of Satisfaction and Dissatisfaction*

Let $x^*$ be the best compromise solution obtained by an individual DM by exploiting the interval outranking relation of Section 4.1.

**Assumption 4 (capacity to set the limiting boundary between classes).** *Let $C_{sat}$ and $C_{dis}$ denote classes "satisfactory" and "unsatisfactory", respectively. Consider $\boldsymbol{\beta} > 0.5$ ($\beta^- > 0.5$) and $\boldsymbol{\lambda} > 0.5$ ($\lambda^- > 0.5$). Taking $x^*$ as reference, the DM is able to set a boundary $B = \{b_1, \ldots b_n\}$ between $C_{sat}$ and $C_{dis}$ fulfilling:*

i. *$f_j(b_k)$ are real numbers, $j = 1, \ldots N$ and $k = 1, \ldots n$;*
ii. *Each $b_k$ ($k = 1, \ldots n$) belongs to $C_{sat}$;*
iii. *For all $b_k$ belonging to $B$, we have $x^* P_r(\boldsymbol{\beta}, \boldsymbol{\lambda}) b_k$;*
iv. *There is no pair $(b_i, b_k)$ such that $b_i P_r(\boldsymbol{\beta}, \boldsymbol{\lambda}) b_k$.*

Let us back to Definition 2, which establishes relations between solutions and the boundary. Replace $\beta$ with the interval number $\boldsymbol{\beta}$. Assumption 4 guarantees the conditions to apply INTERCLASS-nB (see Section 2.4) to making judgments of satisfaction and dissatisfaction.

From Definition 2, it follows that $xS(\boldsymbol{\beta}, \boldsymbol{\lambda})B \Rightarrow not\ (BP_r(\boldsymbol{\beta}, \boldsymbol{\lambda})x)$ and $BP_r(\boldsymbol{\beta}, \boldsymbol{\lambda})x \Rightarrow not\ (xS(\boldsymbol{\beta}, \boldsymbol{\lambda})B)$. If a solution $x$ outranks the limiting boundary $B$, $x$ is assigned to the best class by the "pessimistic" procedure (Definition 3); additionally, since $xS(\boldsymbol{\beta}, \boldsymbol{\lambda})B \Rightarrow not\ (BP_r(\boldsymbol{\beta}, \boldsymbol{\lambda})x)$, $x$ is also assigned to $C_{sat}$ by the "optimistic" procedure (Definition 4) contrarily, if the boundary is preferred to $x$, since $BP_r(\boldsymbol{\beta}, \boldsymbol{\lambda})x \Rightarrow not\ (xS(\boldsymbol{\beta}, \boldsymbol{\lambda})B)$, $x$ will be assigned to the worst class by both assignment rules. If $not\ (xS(\boldsymbol{\beta}, \boldsymbol{\lambda})B$ and $not\ (BP_r(\boldsymbol{\beta}, \boldsymbol{\lambda})x)$ are both fulfilled, then $x$ is assigned to $C_{dis}$ by the "pessimistic" rule and to $C_{sat}$ by the "optimistic" procedure. In such a case, the DM may be doubtful about the class $x$ should be assigned to.

**Assumption 5 (compatibility with INTERCLASS-nB).** *Suppose that $x$ and its pre-image satisfy the constraints imposed by the i-th DM. We assume that the DM is willing to accept the assignment suggested by INTERCLASS-nB.*

As a consequence of Assumption 5, even if $xS(\boldsymbol{\beta}, \boldsymbol{\lambda})B$ or $not\ (BP_r(\boldsymbol{\beta}, \boldsymbol{\lambda})x)$, the nonfulfillment of the constraints vetoes a satisfactory assignment. Based on this assumption, the definitions of what is a satisfactory (non-satisfactory) solution for a DM (who is compatible with the interval outranking model) are straightforward.

**Definition 11.** *Suppose that an individual DM is compatible with the outranking model of Section 4. The DM is said to be satisfied with a solution x iff the following conditions are fulfilled:*

*(a)* *$xS(\boldsymbol{\beta}, \boldsymbol{\lambda})B$*
*(b)* *x and its pre-image satisfy the constraints imposed by the DM.*

**Definition 12.** *Under the same premise of the above definition, the DM is said to be dissatisfied with a solution x iff at least one of the following conditions is fulfilled:*

*(A)*   $BP_r(\boldsymbol{\beta},\boldsymbol{\lambda})x$

*(B)*   *x and/or its pre-image do not satisfy the constraints imposed by the DM.*

**Definition 13.** *The DM is neither satisfied nor dissatisfied with a solution x if the following conditions are all fulfilled:*

1.   *not ($xS(\boldsymbol{\beta},\boldsymbol{\lambda})B$)*
2.   *not ($BP_r(\boldsymbol{\beta},\boldsymbol{\lambda})x$)*
3.   *x and its pre-image satisfy the constraints imposed by the DM.*

It is obvious that the above definitions follow from the INTERCLASS-nB assignment rules and Assumption 5.

Suppose that the individual DM is satisfied with a solution *x*. Let us define the outranking credibility index of *x* with respect to the boundary *B* as follows:

$$\sigma(x, B, \boldsymbol{\lambda}) = max \{\sigma(x, b_k, \boldsymbol{\lambda}), k = 1, \dots \ card(B) \tag{17}$$

The value of $\sigma$ can be interpreted as a measure of how much the DM feels satisfaction with *x*. The following proposition states several consistency properties of the assignments from Definitions 11–13.

**Proposition 2 (consistency properties of assignments).** *Under Assumptions 4 and 5, suppose that x and its pre-image satisfy the constraints imposed by the DM. The following propositions are valid:*

*(a)*   *x is assigned to a single element of the set of classes (satisfactory, unsatisfactory, neither satisfactory nor unsatisfactory).*

*(b)*   *The assignment suggested for x is independent of the assignment of other solutions.*

*(c)*   *The class to which x is assigned by the i-th DM is independent of the assignment made by any other DM.*

*(d)*   *Let y be a feasible solution. Given $\boldsymbol{\lambda}$, if x and y have the same interval outranking credibility indices with respect to the limiting profiles, then they are assigned to the same element of the set of classes (satisfactory, unsatisfactory, neither satisfactory nor unsatisfactory).*

*(e)*   *If there is $b_k \in B$ fulfilling $x = b_k$, then x is assigned to the satisfactory class.*

*(f)*   *If there is $b_k$ such that $xI(\boldsymbol{\beta},\boldsymbol{\lambda})b_k$ and there is no $b_i \in B$ fulfilling $b_iP_r(\boldsymbol{\beta},\boldsymbol{\lambda})x$, then x is assigned to the satisfactory class.*

*(g)*   *If $x = x^*$, then x is assigned to the satisfactory class.*

*(h)*   *Let y be a feasible solution such that $y D(\alpha)x$ $(\alpha \geq \boldsymbol{\beta})$. If x is assigned to the satisfactory class, then y is assigned to the same class.*

**Proof.**

| | |
|---|---|
| Proposition 2(a): | The proof follows from two facts: (i) *x* has to fulfill one of the three Definitions 11–13, and (ii) the fulfillment of one definition excludes fulfillment of another. |
| Proposition 2(b): | The proof is obvious from Definitions 11–13. |
| Proposition 2(c): | The proof is obvious from Definitions 11–13. |
| Proposition 2(d): | The proof is obvious from Definitions 11–13. |
| Proposition 2(e): | $x = b_k$, Assumption 4.i, Remark 2.a, Definition 2.i and Assumption 4.iv $\Rightarrow xS(\boldsymbol{\beta},\boldsymbol{\lambda})B$ *x* is feasible and $xS(\boldsymbol{\beta},\boldsymbol{\lambda})B \Rightarrow x$ is satisfactory for the DM (Definition 11). |
| Proposition 2(f): | $xI(\boldsymbol{\beta},\boldsymbol{\lambda})b_k \Rightarrow xS(\boldsymbol{\beta},\boldsymbol{\lambda})b_k$ (Definition 1.iii) $xS(\boldsymbol{\beta},\boldsymbol{\lambda})b_k$ and there is no $b_i \in B$ such that $b_iP_r(\boldsymbol{\beta},\boldsymbol{\lambda})x \Rightarrow xS(\boldsymbol{\beta},\boldsymbol{\lambda})B$ (Definition 2.i) *x* is feasible and $xS(\boldsymbol{\beta},\boldsymbol{\lambda})B \Rightarrow x$ is satisfactory for the DM (Definition 11). |

| Proposition 2(g): | The proof follows trivially from Assumption 4.iii, Definition 2.i and Definition 11. |
|---|---|
| Proposition 2(h): | $x$ is assigned to $C_{sat} \Rightarrow xS(\boldsymbol{\beta},\boldsymbol{\lambda})B$ from Definition 11 $\Rightarrow \exists\, b_k \in B$ such that $xS(\boldsymbol{\beta},\boldsymbol{\lambda})b_k$ and there is no $b_i \in B$ with $b_i P_r(\boldsymbol{\beta},\boldsymbol{\lambda})x$. |

A. From Proposition 1.*i* and Remark 2.b, $yD(\alpha)x$ ($\alpha \geq \beta$) and $xS(\boldsymbol{\beta},\boldsymbol{\lambda})b_k \Rightarrow yS(\boldsymbol{\beta},\boldsymbol{\lambda})b_k$.

B. There is no $b_i \in B$ with $b_i P_r(\boldsymbol{\beta},\boldsymbol{\lambda})x$ and $y\, D(\alpha)x$ ($\alpha \geq \beta$) $\Rightarrow$ There is no $b_i \in B$ with $b_i P_r(\boldsymbol{\beta},\boldsymbol{\lambda})y$ counter-reciprocal of Proposition 1.ii.

$\square$

Combining (A) and (B), we have $yS(\boldsymbol{\beta},\boldsymbol{\lambda})B$ (Definition 2.*i*) $\Rightarrow y$ is assigned to $C_{sat}$ (Definition 11). The proof is finished.

It should be noticed that the first condition in Assumption 4 (real values in the limiting actions) is required to prove Property 2.e and thus achieving consistency with Condition *ii* in Assumption 5. From Propositions 2.e and 2.f, it follows that a solution only slightly different from any $b_k \in B$ should be considered as satisfactory by the DM.

## 5. Model for a DM Whose Preferences Are Compatible with a Weighted-Sum Function

### 5.1. The Preference Model

In this section, we suppose that the *i*-th DM's preferences are compatible with a simple interval weighted-sum value function of the form:

$$\boldsymbol{U} = \Sigma w_j f_j \ (j = 1, \dots N) \tag{18}$$

In Equation (18), $\boldsymbol{U}$ is an interval number that is calculated through the arithmetic operations defined in Section 2.3. We refer to a generic DM, and for simplicity, we avoid the use of the subscript "*i*".

**Assumption 6 (modeling arbitrariness and ill-definition of parameters).** *The DM is willing to set a weighted-sum function model in terms of weights and objective function values. With the purpose of modeling imperfect information, those magnitudes will be considered as interval numbers.*

This assumption was implicitly made by Liesio et al. (2007, 2008) and Fliedner and Liesio (2016) in the frame of project portfolio optimization.

**Remark 4.** *Let us consider a pair (x,y) in the objective space of Problem 11. The conditions $P(\boldsymbol{U}(x) \geq \boldsymbol{U}(y)) > 0.5$ or xD(0.5)y may not suffice to guarantee a credible preference favoring x over y. However, there should be a credibility threshold $\alpha > 0.5$ such that $P(\boldsymbol{U}(x) \geq \boldsymbol{U}(y)) \geq \alpha$ and/or xD($\alpha$)y are good arguments to justify a credible preference.*

**Definition 14 ($\alpha$-preference).** *Given a pair (x,y) in the objective space of Problem 11 and $\alpha$ a real number greater than 0.5, we say that x is $\alpha$-preferred to y iff at least one of the following conditions is fulfilled:*

*a.   $P(\boldsymbol{U}(x) \geq \boldsymbol{U}(y)) \geq \alpha$*
*b.   xD($\alpha$)y*

### 5.2. Identifying the Best Compromise Solution to Problem 11 with the Functional Preference Model

We follow here a rather similar way as in Section 4.2 to make operational Assumption 2 in the frame of the functional model of preferences.

**Definition 15 ($\alpha$ non-strictly outranked solution).** *A solution $x \in O$ will be called $\alpha$ non-strictly outranked iff there is no $y \in O$ such that y is $\alpha$-preferred to x. The set of these solutions is called the $\alpha$ non-strictly outranked frontier of Problem 11.*

**Remark 5.**

I. *With an appropriate value of α, a non-strictly outranked solution from Definition 15 fulfills Condition A of Definition 7, that is, the first necessary condition to be the best compromise solution of Problem 11.*

II. *Condition B of Definition 7 is verified through a value strength measure on the non-strictly outranked frontier. This measure is defined as VS(x) = card {y ∈ NSF such that x is 0.5-preferred to y}, where NSF denotes the α non-strictly outranked frontier.*

III. *Several solutions can fulfill the necessary conditions of Definition 7. The final best compromise should be one of the solutions with the highest measure VS.*

Under the interval value function approach, Problem 11 can be solved through the following steps:

1. The individual DM (perhaps helped by a decision analyst) sets the interval weights in Equation 18 and the $α$ value.
2. An optimization algorithm is used to identify the $α$ non-strictly outranked frontier.
3. The set of solutions that fulfill Definition 10 is identified.
4. The DM selects the best compromise solution according to Remark 5.III.

Similarly to Section 4, the appropriate optimization algorithm in the second step depends on the characteristics of the problem. For combinatorial problems like project portfolio optimization, we can exploit the $α$-preference defined on $O × O$ with only slight changes with respect to the I-NOSGA proposed in [38] (see Appendix A for a detailed description).

*5.3. Making Judgments of Satisfaction and Dissatisfaction with the Functional Model*

We will present here an idea that is, to a certain extent, similar to the one on which INTERCLASS-nB and the proposal in Section 4.3 are inspired. If the DM is capable of setting a limiting boundary between the two classes, all feasible solutions preferred to the boundary should be assigned to the best class and vice versa.

Formally: Let $x^*$ be the best compromise solution obtained by the individual DM following the procedure exposed in the previous section.

**Assumption 7 (capacity to set the limiting boundary between classes).** *Let $C_{sat}$ and $C_{dis}$ denote classes "satisfactory" and "unsatisfactory", respectively. Consider a sufficiently high value of the credibility threshold α. Taking $x^*$ as reference, the DM is able to set a limiting boundary B= {b₁, . . . bₙ} between $C_{sat}$ and $C_{dis}$ fulfilling:*

i. *For all $b_k$ belonging to B, we have $x^*$ is α-preferred to $b_k$;*
ii. *There is no pair $(b_i, b_k)$ in B × B such that $b_i$ is α-preferred to $b_k$;*
iii. *For all $b_k$ belonging to B, the DM hesitates about its assignment.*

**Definition 16 (α-preference between a solution and the boundary).**

(a) *x is α-preferred to B iff there is $b_k ∈ B$ such that x is α-preferred to $b_k$ and there is no $b_i ∈ B$ such that $b_i$ is α-preferred to x.*

(b) *The boundary B is α-preferred to x iff there is $b_k ∈ B$ such that $b_k$ is α-preferred to x and there is no $b_i ∈ B$ such that x is α-preferred to $b_i$.*

**Definition 17.** *Consider $x ∈ O$ and $b_k ∈ B$. The credibility index of the statement "x is preferred to $b_k$" is defined as $σ_P (x, b_k)$ = Maximum χ fulfilling at least one of the following conditions:*

a. *$P(U(x) ≥ U(b_k)) = χ$*
b. *$xD(χ)b_k$*

The credibility index $σ_P (x, B)$ of the statement "x is preferred to the boundary B" is defined as:

$$σ_P (x, B) = Max \{σ_P (x, b_k)\}, k = 1, \ldots card (B) \tag{19}$$

$\sigma_P(x, B)$ can be interpreted as the degree of credibility of the statement "$x$ is preferred to the boundary by the $i$-th DM".

**Definition 18.** *Suppose that an individual DM is compatible with the weighted-sum function model and Assumptions 6 and 7. The DM is said to be satisfied with a solution x iff the following conditions are fulfilled:*

-      *x is α-preferred to the Boundary B;*
-      *x and its pre-image fulfill the constraints imposed by the DM.*

**Definition 19.** *Suppose that an individual DM is compatible with the weighted-sum function model and Assumptions 6 and 7. The DM is said to be dissatisfied with a solution x if at least one of the following conditions is fulfilled:*

-      *B is α-preferred to x;*
-      *x and/or its pre-image do not fulfill the constraints imposed by the DM.*

**Definition 20.** *The DM is neither satisfied nor dissatisfied with a feasible solution x iff the following conditions are held:*

-      *x is not α-preferred to the Boundary B;*
-      *B is not α-preferred to x.*

**Proposition 3 (consistency properties of assignments).** *Suppose that x is feasible. The following propositions are valid:*

- *(a)*   *x is assigned to a single element of the set of classes (satisfactory, unsatisfactory, neither satisfactory nor unsatisfactory).*
- *(b)*   *The assignment suggested for x is independent of the assignment of other solutions.*
- *(c)*   *The class to which x is assigned to by the i-th DM is independent of the assignment made by any other DM.*
- *(d)*   *If there is $b_k \in B$ fulfilling $x = b_k$, then x is neither satisfactory nor unsatisfactory.*
- *(e)*   *If $x = x^*$, then x is assigned to the satisfactory class.*
- *(f)*   *Suppose that x is α-preferred to all $b_k \in B$. Let y be a feasible solution such that $yD(\alpha)x$. Then y is assigned to the satisfactory class.*

**Proof.**

| | |
|---|---|
| Proposition 3(a): | The proof is obvious from Definitions 16, 18, 19, and 20. |
| Proposition 3(b): | The proof is obvious from Definitions 16, 18, 19, and 20. |
| Proposition 3(c): | The proof is obvious from Definitions 16, 18, 19, and 20. |
| Proposition 3(d): | From the way in which $B$ is built (Assumption 7), there is no $b_i \in B$ such that $b_k$ is *α-preferred* to $b_i$ or $b_i$ is *α-preferred* to $b_k$. Then, $b_k$ is not *α-preferred* to $B$ and $B$ is not *α-preferred* to $b_k$ (Definition 16). From Definition 20, it follows that $b_k$ is neither satisfactory nor unsatisfactory. |
| Proposition 3(e): | The proof follows from Definitions 16 and 18 and the way in which the limiting boundary is built (Assumption 7). |
| Proposition 3(f): | It is evident that $y\,D(\alpha)x$ and $P(\mathbf{U}(x) \geq \mathbf{U}(b_k)) \geq \alpha \Rightarrow P(\mathbf{U}(y) \geq \mathbf{U}(b_k)) \geq \alpha$. In addition $yD(\alpha)x$ and $xD(\alpha)b_k \Rightarrow yD(\alpha)b_k$ from transitivity of dominance (Proposition 1.v). Hence, $y$ is *α-preferred* to the Boundary $B$ (Definitions 14 and 16). From Definition 18, $y$ is assigned to the satisfactory class. The proof is finished. |

        □

**Remark 6.**

*(a)    Unlike the proposal in Section 4, the solutions in the limiting boundary do not belong to any class. Objectives of these solutions may be described by interval numbers, what gives more flexibility to the DM and may reduce their cognitive effort.*

*(b)    Proposition 3.d is consistent with Condition iii of Assumption 7. It follows that a solution only slightly different from any $b_k \in B$ should be considered neither satisfactory nor unsatisfactory by the DM.*

## 6. Summary of the Method

The models and methods described in Sections 4 and 5 are alternatives. The DMs may select the model of preferences they feel more compatible with. Each DM sets their own model parameters, solves their own Problem 11, identifies their best compromise, and defines their limiting boundary according to Assumption 4 or Assumption 7. Then, it is possible to evaluate the number of satisfied and dissatisfied DMs, exploring the decision variable space to solve Problem 1. Since the preference models and their parameters differ from two disjoint subsets of the whole group, the consensus reaching process should be carried in different subgroups.

The proposal maintains its validity even if the DMs face different optimization problems (with different objectives and constraints) but with common decision variables ranging in the intersection of the feasible regions of each DM optimization problem. The main reasons for such a general performance come from Propositions 2.c and 3.c and the independent way in which each DM sets their limiting boundary and defines what a satisfactory (unsatisfactory) solution is. The specific characteristics, preferences, and judgments from each DM influence on determining whether particular solutions are satisfactory or not for the respective group member. The collective final decision comes from optimizing $(N_{sat}, N_{dis})$ in Problem 1, but the calculation of these objectives is made by aggregating the independent assignments of all the DMs.

It should be underlined that the proposal is, to a great extent, robust with respect to malicious intents of manipulation from some group members. To analyze this issue, suppose that some DMs are interested in manipulating the final solution. To achieve this, they should try (providing insincere information), to increase $N_{sat}(z^*)$ and reduce $N_{dis}(z^*)$ in the points $z^*$ where they are really satisfied, and to decrease $N_{sat}(z^{**})$ and increase $N_{dis}(z^{**})$ in the points $z^{**}$ where they are really dissatisfied. The information required from them is basically of three kinds: (i) parameters of their preference model; (ii) solutions describing the limiting boundary between classes "satisfactory" and "unsatisfactory"; (iii) their constraints and objective values. This information determines if the DMs are satisfied or dissatisfied and counted in $N_{sat}$ or $N_{dis}$. If they report insincere information on any of the three previous aspects, solutions that are really satisfactory for them could become unsatisfactory according to the model, and vice versa. This does not contribute to reaching a final solution really satisfactory for the DMs that provide insincere information. In addition, each DM should be impeded to know the limiting boundaries elicited by other DMs. Under this restriction, the *i*-th DM cannot evaluate solutions from the point of view of other DMs, being thus unable to determine what insincere information should provide for maliciously influencing other opinions. Therefore, during collective discussions, the *i*-th DM should defend their real preferences and beliefs and try to convince others.

The method for finding the maximum consensus can be summarized as steps:

1.    Helped by a decision analyst/moderator, the DMs select the model of multi-criteria preferences that they consider as more appropriate. The group is separated into two disjoint subgroups in correspondence to the model of preferences that were chosen by each DM.

2.    In each subgroup, under the guidance of the moderator, the DMs bring their positions closer. They exchange opinions about the objective functions to consider, the objective values, the related model's parameter values, levels of conservatism, and constraints.

3.  Each group member sets their multi-objective optimization problem (Problem 11) and their model's parameter values. Interval numbers can be used according to Assumptions 1, 3, and 6.

4.  According to Assumption 2, each group member obtains their best compromise solution by solving Problem 11.

5.  Each DM sets their limiting boundary in correspondence to Assumption 4 (when the DM is compatible with the outranking model) and Assumption 7 (for DMs compatible with the functional model).

6.  Applying the classification models of Sections 4 and 5, the moderator finds non-dominated solutions ($N_{sat}$, $N_{dis}$) of Problem 1.

7.  **If there is no solution of good agreement**, further discussions in each subgroup are needed to close divergent beliefs, preference parameters, and constraint settings. We need to update these data.

8.  The DMs should judge whether, given the updated data, they want to modify their limiting boundary. In the case of "yes", restart the process in Step 5. In the case of "no", restart the process in Step 6. **If a good consensus ($N_{sat}$, $N_{dis}$)\* is found, then:**

9.  If the pre-image of ($N_{sat}$, $N_{dis}$)\* is a single point in the decision variables space, this point corresponds to the best consensus, and the process finishes. **Else:**

10. Apply some additional criterion to select a single pre-image of ($N_{sat}$, $N_{dis}$)\*. The process finishes.

The method is based on seven assumptions. Table 1 summarizes the set of assumptions and their role within the approach.

**Table 1.** Summary of assumptions.

|  | It's Subject | Allows | Related to |
|---|---|---|---|
| Assumption 1 | Interval numbers as model of imprecisions | Modeling imprecision | Steps 1–2 |
| Assumption 2 | Capacity to identify the best compromise | Identifying best compromises | Step 4 |
| Assumption 3 | Compatibility with an outranking model | Preference modeling | Step 1 |
| Assumption 4 | Capacity to set the limiting boundaries related to the outranking model | Identifying limiting boundaries | Step 5 |
| Assumption 5 | Compatibility with INTERCLASS-nB | Assigning solutions to classes of satisfaction/dissatisfaction | Step 6 |
| Assumption 6 | Compatibility with an interval value function | Preference modeling | Step 1 |
| Assumption 7 | Capacity to set the limiting boundaries related to the value function model | Identifying limiting boundaries | Step 5 |

All the DMs should accept Assumptions 1 and 2. DMs who are compatible with the outranking model should agree to Assumptions 3–5, whereas the other DMs should accept Assumptions 6 and 7.

**Remark 7.**

i.  *In the case of project portfolio optimization, the computational cost depends mainly on Step 4. The computational complexity of this step is linear with respect to the number of applicant projects (see the description of the I-NOSGA algorithm in Appendix A).*

ii. *Handling group interactions in Steps 1, 2, and 7 is the main difficulty to extend the proposal to problems with many DMs. In such problems, Steps 3, 4, 5, and 8 should be performed by each DM in an independent and parallel way. Parallel processing in Step 4 is strongly recommended. Steps 9 and 10 are independent of the number of group members. The computational effort in Step 6 is, at most, proportional to the number of DMs (see Appendix C). Therefore, with some modifications in Steps 1, 2, and 7, the proposal can be used in large-scale GDM-MOPs.*

iii. *In Step 7, in order to accept a solution as a good consensus, the group may agree previously appropriate values of $N_{sat}$ and $N_{dis}$ to represent what a strong majority and a weak minority mean, respectively.*

iv. *In Step 10, there could be several (even many) pre-images of the best consensus $(N_{sat}, N_{dis})^*$. To choose a single one, the group and/or its moderator can use different points of view (e.g., impacts of the solutions, resource consumption, who are the satisfied and dissatisfied DMs, number of supported projects in case of portfolio optimization, etc.). Perhaps the most elegant way is a logical approach based on the outranking credibility index of a solution with respect to the limiting boundary (see Equations (17) and (19)).*

Let us denote as $z_{G1}, \dots, z_{GL}$ the points in the decision variable space, which are all pre-image of the best consensus $(N_{sat}, N_{dis})^*$. Let $C_{ag}$ denote the agreeing coalition (the set of group members who are satisfied with the image of $z_{Gk}$). In the paragraph below, we use "*i*" (respectively "*j*") for the DMs who are compatible with the outranking (resp. weighted sum functional) model.

If we denote as $Z_{Gki}$ (resp. $Z_{Gkj}$) the image of $z_{Gk}$ in the original objective space $O_i$ (resp. $O_j$), $\sigma_i(Z_{Gki}, B_i, \lambda_i)$ (resp. $\sigma_{Pj}(Z_{Gkj}, B_j)$) can be interpreted as the degree of truth of the predicate "the *i*-th DM (resp. the *j*-th DM) considers that $Z_{Gki}$ (resp. $Z_{Gkj}$) outranks the limiting boundary $B_i$ (resp. $B_j$)". Then, the degree of truth of the predicate "all the DMs belonging to the agreeing coalition consider that $Z_{Gki}$ (resp. $Z_{Gkj}$) outranks the related limiting boundary" can be calculated as the conjunction of all the values $\sigma_i(Z_{Gki}, B_i, \lambda_i)$ and $\sigma_{Pj}(Z_{Gkj}, B_j)$, where "*i*" (resp. "*j*") is the index of the *i*-th DM (resp. the *j*-th DM) within the agreeing coalition. Such a truth value is calculated as:

$$\mu_{sat}(z_{Gk}) = min\{\sigma_i(Z_{Gki}, B_i, \lambda_i), \sigma_{Pj}(Z_{Gkj}, B_j)\} \qquad (20)$$
$$i, j \in C_{ag}$$

**Remark 8.** *Although the proposal exhibits remarkable advantages in comparison with previous approaches to group multi-objective optimization, it can be still criticized from the following points:*

- To set the limiting boundaries could be a hard cognitive task for DMs; it would be more complex in large scale problems.
- The bi-objective measure of collective satisfaction/dissatisfaction does not contain information about which DMs are strongly (dis)satisfied. This information can be important to discriminate among non-dominated solutions of Problem 1. Perhaps the multi-criteria ordinal classification method should take into account more classes of satisfaction/dissatisfaction, but this would require much more cognitive effort from the DMs in defining more limiting boundaries.
- The role of the moderator is crucial in choosing the final consensus decision among the non-dominated solutions of Problem 1. A model of consensus agreed by the group would be welcome. Such a model should aggregate the information about satisfaction/dissatisfaction, thus helping to make the final choice among non-dominated solutions to Problem 1.

## 7. An Illustrative Example of Project Portfolio Optimization

In this section, we will apply both models of multi-criteria preferences to the many-objective project portfolio optimization problem addressed by Balderas et al. (2019). For simplicity of our illustrative purpose and with a little loss of generality, we consider that all the DMs are compatible with the same model (interval outranking or interval weighted-sum function).

The case study involves 100 applicant social projects whose impacts are described by nine objectives. We suppose that all the DMs agree on the same set of objective functions. The description of the project impacts and their budget requirements are shown in: https://www.dropbox.com/s/t80u9kbdcub6jua/Projects%20Description.pdf?dl=0 (accessed on 12 May 2021).

The objectives are the number of beneficiaries identified per social group and the level of impact of this benefit. The group is integrated by 10-equally important DMs, who may interact several rounds until reaching consensus. Consistently with Assumption 1, let us suppose that in the first interaction, the moderator achieved an initial agreement of the DMs on the project impacts and budget requirements, given as interval numbers. These intervals, specified in dollars, represent the variability of the DM's judgments and beliefs. Derived from the first interactive round, the feasible region, which is defined by the available budget, is set as the interval [240,260] million dollars.

### 7.1. Solution When All the DMs Accept the Interval Outranking Model and Its Assumptions

Based on Assumption 3, the parameter values of the model (specified by the group members) are shown in Table A2 (See Appendix B). Weights and vetoes are the same as in [38].

Problem 11 was solved by each group member through the procedure explained in Section 4.2 and the evolutionary algorithm in Appendix C with the parameters given in Table A2a–c. The best compromise solution was identified by each DM (Assumption 2), as shown in Table 2. The conditions under which our computer experiments were carried out are the following: (1) the algorithms were implemented using the Java programming language and run on a computer with Intel Core i7-6700HQ 2.6 GHz CPU, 8 GB of RAM, and Windows 10 Pro operating system; and (2) the solutions were obtained from 30 independent runs. The parameter values for the algorithms were tuned through a covering array experimental design [58]. The involved factors were population size, number of evaluations of the objective function vector and mutation probability, with levels of {50, 100, 200}, {10,000, 100,000, 200,000} and {0.01, 0.02, 0.05}, respectively. The results from the fine-tuning process indicated as the best configuration the following values: Population size = 100, number of evaluations = 100,000, and mutation probability = 0.02. The time required by the extended I-NOSGA to solve Problem 11 per group member was of 153 min and 56 s.

Taking their best compromise as an ideal reference, the DMs selects their associated limiting boundary between $C_{sat}$ and $C_{dis}$. Each boundary is composed of three solutions that fulfill the conditions in Assumption 4. The boundaries are described in Table A3 in Appendix B.

There is $(\beta,\lambda)$-preference favoring the best compromise in its comparison with the solutions in the limiting boundary (Condition *iii* in Assumption 4), but there is no $\alpha$-dominance.

Now, for identifying the best consensus, we should solve Problem 1. We use MOEA/D, whose brief description can be found in Appendix C. This algorithm found two solutions in the decision variable space, both fulfilling $N_{sat} = 9$ and $N_{dis} = 0$. In the objective space, such solutions are given in Table 3. The time consumed by MOEA/D to solve Problem 1 was 35 min and 59 s.

There is no dissatisfaction with such solutions. All the DMs are satisfied except a single one who is neither satisfied nor dissatisfied. It is a high consensus level that may be accepted as the final collective decision. However, we suppose that the group moderator wishes to carry on a consensus reaching step to close even more the group judgments and beliefs.

Suppose that, under the guidance from the moderator; the group members make closer criterion weights, veto thresholds, credibility thresholds $\alpha$, $\beta$, and $\xi$, and the required budget per project. We made a computer simulation of this consensus round as follows:

The procedure in Algorithm 1 is applied for Parameter $p$ in the set MP = {lower bound of $w_j$ ($j = 1, \dots, 9$), upper bound of $w_j$ ($j = 1, \dots, 9$), lower bound of $v_j$ ($j = 1, \dots, 9$), upper bound of $v_j$ ($j = 1, \dots, 9$), lower bound of $\beta$, and upper bound of $\beta$, $\alpha$, $\xi$}.

**Algorithm 1.** Consensus Round Simulation

For each $p$ in MP
Let $p_i$ be the value of Parameter $p$ set by the $i$-th DM.
Let $p_a$ denote the average value of $p$ on the set of DMs.
Repeat from $i = 1$ to 10
Calculate $d = p_i - p_a$
If $d > 0$, update $p_i$ as $p_i - d/2$
If $d < 0$, update $p_i$ as $p_i + |d/2|$
If $d = 0$, then $p_i$ keeps its value
End of Repeat
End of For

Applying Algorithm 1, the distance between the previous value of $p$ from the $i$-th DM and the average value of such a parameter on the set of DMs is reduced by half. The interval budget required by projects is updated from their original values (see https://www.dropbox.com/s/t80u9kbdcub6jua/Projects%20Description.pdf?dl=0 (accessed on 12 May 2021)) in such a way that the new interval number has the same middle point but whose length is reduced by half. The updated weights, vetoes, and credibility thresholds are provided by Table A4a–c, and the updated budget per project is shown in Table A5 (see Appendix B). $\lambda$ remains the same as in Table A2c.

With the updated values, again, MOEA/D is used in solving Problem 1. Since preferences, attitudes facing imperfect knowledge, and judgments from the DMs are closer than before; the optimization problem is now more relaxed. The ideal consensus $N_{sat} = 10$, $N_{dis} = 0$ is achieved, with many pre-images in the decision variable space. In this example, the solutions satisfying the ideal consensus are ordered following the values of $\mu_{sat}$ in Equation (20). The first ranked solutions are shown in Table 4.

The values $\sigma_i(Z_{Gki}, B_i, \lambda_i)$ from Equation 17 are provided by Table A6 (see Appendix B). These values represent measures of the level of satisfaction for each group member.

**Table 2.** Best compromise solution for each DM (the *i*-th row corresponds to the *i*-th DM).

| O1 | | O2 | | O3 | | O4 | | O5 | | O6 | | O7 | | O8 | | O9 | |
|---|---|---|---|---|---|---|---|---|---|---|---|---|---|---|---|---|---|
| 1,234,925 | 1,337,795 | 995,015 | 1,077,935 | 1,417,065 | 1,535,135 | 929,930 | 1,007,440 | 1,507,725 | 1,633,405 | 1,211,330 | 1,312,210 | 1,925,345 | 2,085,795 | 1,365,315 | 1,479,045 | 1,658,795 | 1,796,975 |
| 1,259,905 | 1,364,855 | 975,365 | 1,056,625 | 1,474,745 | 1,597,625 | 959,470 | 1,039,430 | 1,480,450 | 1,603,850 | 1,279,330 | 1,385,880 | 1,904,455 | 2,063,165 | 1,486,095 | 1,609,895 | 1,633,700 | 1,769,780 |
| 1,239,515 | 1,342,765 | 1,024,090 | 1,109,430 | 1,449,225 | 1,569,975 | 924,815 | 1,001,875 | 1,643,530 | 1,780,530 | 1,255,190 | 1,359,740 | 2,009,685 | 2,177,205 | 1,358,450 | 1,471,650 | 1,661,035 | 1,799,395 |
| 1,189,995 | 1,289,115 | 1,060,215 | 1,148,545 | 1,425,755 | 1,544,545 | 938,105 | 1,016,275 | 1,600,580 | 1,734,010 | 1,312,095 | 1,421,375 | 1,984,560 | 2,149,970 | 1,370,165 | 1,484,325 | 1,610,035 | 1,744,125 |
| 1,215,945 | 1,317,235 | 1,010,400 | 1,094,590 | 1,429,165 | 1,548,235 | 927,465 | 1,004,755 | 1,634,045 | 1,770,235 | 1,234,430 | 1,337,260 | 1,992,975 | 2,159,115 | 1,383,445 | 1,498,725 | 1,647,005 | 1,784,205 |
| 1,169,800 | 1,267,240 | 1,015,325 | 1,099,915 | 1,442,685 | 1,562,875 | 962,570 | 1,042,800 | 1,586,845 | 1,719,105 | 1,211,310 | 1,312,200 | 2,012,250 | 2,179,980 | 1,445,540 | 1,565,980 | 1,576,135 | 1,707,435 |
| 1,211,085 | 1,311,975 | 1,026,425 | 1,111,955 | 1,394,220 | 1,510,390 | 910,105 | 985,945 | 1,588,335 | 1,720,725 | 1,216,135 | 1,317,415 | 1,948,120 | 2,110,500 | 1,358,375 | 1,471,545 | 1,663,015 | 1,801,545 |
| 1,245,805 | 1,349,585 | 969,110 | 1,049,850 | 1,471,245 | 1,593,835 | 946,720 | 1,025,610 | 1,538,740 | 1,667,020 | 1,231,455 | 1,334,015 | 1,820,550 | 1,972,310 | 1,392,165 | 1,586,475 | 1,596,360 | 1,729,340 |
| 1,139,045 | 1,233,925 | 1,034,565 | 1,120,765 | 1,377,385 | 1,492,165 | 925,280 | 1,002,380 | 1,507,990 | 1,633,660 | 1,215,730 | 1,317,000 | 1,903,360 | 2,061,980 | 1,429,490 | 1,548,590 | 1,584,005 | 1,715,955 |
| 1,262,100 | 1,367,230 | 1,030,870 | 1,116,770 | 1,385,135 | 1,500,545 | 931,350 | 1,008,950 | 1,629,665 | 1,765,505 | 1,201,845 | 1,301,955 | 1,932,615 | 2,093,685 | 1,392,655 | 1,508,695 | 1,628,245 | 1,763,875 |

**Table 3.** Best consensus solution to Problem 1 with $N_{sat} = 9$ and $N_{dis} = 0$.

| O1 | | O2 | | O3 | | O4 | | O5 | | O6 | | O7 | | O8 | | O9 | |
|---|---|---|---|---|---|---|---|---|---|---|---|---|---|---|---|---|---|
| 1,199,230 | 1,299,120 | 1,038,945 | 1,125,525 | 1,431,025 | 1,550,265 | 909,215 | 984,995 | 1,546,755 | 1,675,685 | 1,243,915 | 1,347,515 | 1,939,700 | 2,101,360 | 1,355,805 | 1,468,735 | 1,647,945 | 1,785,225 |
| 1,210,965 | 1,311,855 | 1,048,275 | 1,135,635 | 1,411,335 | 1,528,925 | 908,875 | 984,625 | 1,524,965 | 1,652,065 | 1,243,445 | 1,346,995 | 1,955,170 | 2,118,120 | 1,346,425 | 1,458,585 | 1,633,100 | 1,769,130 |

**Table 4.** Best ranked solutions to Problem 1 with $N_{sat} = 10$ and $N_{dis} = 0$ (after the consensus round).

| O1 | | O2 | | O3 | | O4 | | O5 | | O6 | | O7 | | O8 | | O9 | | $\mu_{sat}$ |
|---|---|---|---|---|---|---|---|---|---|---|---|---|---|---|---|---|---|---|
| 1,283,185 | 1,390,075 | 1,045,960 | 1,133,110 | 1,468,445 | 1,590,815 | 924,670 | 1,001,730 | 1,568,155 | 1,698,875 | 1,253,555 | 1,357,965 | 1,922,775 | 2,083,025 | 1,429,465 | 1,548,565 | 1,646,745 | 1,783,915 | 0.8345 |
| 1,219,040 | 1,320,580 | 1,041,655 | 1,128,455 | 1,471,845 | 1,594,475 | 922,825 | 999,735 | 1,587,925 | 1,720,275 | 1,291,315 | 1,398,885 | 2,028,640 | 2,197,730 | 1,431,800 | 1,551,090 | 1,640,255 | 1,776,895 | 0.8068 |
| 1,241,255 | 1,344,665 | 997,255 | 1,080,345 | 1,491,285 | 1,615,545 | 922,435 | 999,305 | 1,513,260 | 1,639,390 | 1,301,545 | 1,409,955 | 1,949,135 | 2,111,575 | 1,426,035 | 1,544,825 | 1,655,110 | 1,792,980 | 0.7984 |
| 1,222,780 | 1,324,630 | 1,053,970 | 1,141,790 | 1,463,215 | 1,585,125 | 925,105 | 1,002,195 | 1,644,730 | 1,781,850 | 1,262,320 | 1,367,460 | 1,976,335 | 2,141,065 | 1,357,765 | 1,470,895 | 1,642,285 | 1,779,075 | 0.7910 |
| 1,257,785 | 1,362,565 | 1,043,480 | 1,130,430 | 1,462,865 | 1,584,755 | 913,840 | 989,990 | 1,639,715 | 1,776,395 | 1,245,910 | 1,349,680 | 2,005,805 | 2,172,985 | 1,392,880 | 1,508,940 | 1,641,615 | 1,778,365 | 0.7880 |

**Table 5.** Best compromise solution for each DM (the *i*-th row corresponds to the *i*-th DM).

| O1 | | O2 | | O3 | | O4 | | O5 | | O6 | | O7 | | O8 | | O9 | |
|---|---|---|---|---|---|---|---|---|---|---|---|---|---|---|---|---|---|
| 1,172,035 | 1,269,655 | 935,955 | 1,013,925 | 1,293,390 | 1,401,160 | 921,260 | 998,020 | 1,703,365 | 1,845,315 | 1,059,975 | 1,148,295 | 1,831,405 | 1,984,095 | 1,366,425 | 1,480,285 | 1,420,215 | 1,538,515 |
| 1,210,295 | 1,311,165 | 989,555 | 1,072,025 | 1,322,445 | 1,432,625 | 899,420 | 974,370 | 1,607,385 | 1,741,335 | 1,165,840 | 1,262,970 | 1,915,940 | 2,075,640 | 1,451,480 | 1,572,410 | 1,478,550 | 1,601,720 |
| 1,169,955 | 1,267,395 | 1,012,675 | 1,097,065 | 1,366,235 | 1,480,075 | 871,970 | 944,640 | 1,771,940 | 1,919,620 | 1,191,690 | 1,290,980 | 1,874,150 | 2,030,360 | 1,469,130 | 1,591,510 | 1,433,435 | 1,552,855 |
| 1,202,270 | 1,302,430 | 1,006,320 | 1,090,160 | 1,393,975 | 1,510,105 | 897,320 | 972,100 | 1,566,755 | 1,774,835 | 1,163,860 | 1,260,850 | 1,914,190 | 2,073,740 | 1,462,680 | 1,584,530 | 1,479,985 | 1,603,305 |
| 1,210,395 | 1,311,205 | 1,005,315 | 1,089,095 | 1,379,125 | 1,494,045 | 843,045 | 913,305 | 1,668,500 | 1,807,560 | 1,184,000 | 1,282,630 | 1,869,795 | 2,025,655 | 1,465,640 | 1,587,720 | 1,392,025 | 1,507,935 |
| 1,134,520 | 1,228,990 | 1,002,865 | 1,086,435 | 1,283,820 | 1,390,790 | 864,205 | 936,225 | 1,667,285 | 1,806,245 | 1,098,580 | 1,190,120 | 1,872,725 | 2,028,815 | 1,356,185 | 1,469,155 | 1,448,755 | 1,569,405 |
| 1,172,035 | 1,269,655 | 935,955 | 1,013,925 | 1,293,390 | 1,401,160 | 921,260 | 998,020 | 1,703,365 | 1,845,315 | 1,059,975 | 1,148,295 | 1,831,405 | 1,984,095 | 1,366,425 | 1,480,285 | 1,420,215 | 1,538,515 |
| 1,142,135 | 1,237,275 | 930,485 | 1,008,005 | 1,331,435 | 1,442,355 | 876,760 | 949,810 | 1,619,515 | 1,826,155 | 1,147,440 | 1,243,030 | 1,905,355 | 2,064,205 | 1,465,220 | 1,587,270 | 1,451,000 | 1,571,890 |
| 1,157,660 | 1,254,110 | 981,860 | 1,063,670 | 1,363,505 | 1,477,115 | 890,910 | 965,140 | 1,535,575 | 1,663,575 | 1,182,040 | 1,280,510 | 1,775,935 | 1,923,955 | 1,396,465 | 1,512,805 | 1,434,775 | 1,554,305 |
| 1,164,705 | 1,261,715 | 962,750 | 1,042,940 | 1,275,670 | 1,381,990 | 860,665 | 932,405 | 1,605,355 | 1,739,175 | 1,141,455 | 1,236,555 | 1,822,560 | 1,974,440 | 1,452,580 | 1,573,600 | 1,411,810 | 1,529,430 |

### 7.2. Solution When All the DMs Accept the Interval Dum Function Model and Its Assumptions

The criterion weights in this model are in general different from the weights in an outranking model. Based on Assumption 6, for simplicity and without loss of generality, we set in this Section the same values provided by Table A2a. The model of Section 5 requires credibility thresholds $\alpha$ and $\xi$. These are set as in Table A2c. Specifications about projects and requirements are the same as in https://www.dropbox.com/s/t80u9kbdcub6 jua/Projects%20Description.pdf?dl=0 (accessed on 12 May 2021).

Problem 11 was solved by each group member through the procedure explained in Section 5.2 and the evolutionary algorithm in Appendix C with the parameters given in Table A2a,c. According to Assumption 2, the best compromise solution for each DM was determined, as shown in Table 5. Under the same conditions defined in the previous experiments, the extended I-NOSGA algorithm required 37 min and 47 s per group member to solve Problem 11.

With their best compromise as an ideal reference, the DMs identify their associated limiting boundary between $C_{sat}$ and $C_{dis}$. Each boundary is composed of three solutions that fulfill the conditions in Assumption 7. The boundaries are detailed in Table A7 in Appendix B.

Once the limiting boundaries have been set, the evaluation of functions $N_{sat}$ and $N_{dis}$ is straightforward through Definitions 18–20. Using MOEA/D (see Appendix C), a single non-dominated solution of Problem 1 was found, with $N_{sat} = 6$, $N_{dis} = 0$. In the original nine-objective space, this solution is given in Table 6.

As in Section 7.1, suppose that, under the guidance from the moderator, the group members make closer criterion weights, credibility thresholds $\alpha$ and $\xi$, and the required budget per project. Our computer simulation of this consensus round, similarly to Section 7.1, is performed through Algorithm 1. Here, the procedure in Algorithm 1 is applied for Parameter $p$ in the set $MP$ = {lower bound of $w_j$ ($j$ = 1, . . . , 9), upper bound of $w_j$ ($j$ = 1, . . . , 9), $\alpha$, $\xi$}. Since the simulation algorithm is identical, the updated parameters are the same as in Table A4a,c. The interval budget required by projects is updated identically to the way followed in Section 7.1 (see the updated values in Table A5, Appendix B).

Solving Problem 1, again, MOEA/D identified many solutions in the decision variable space, which satisfies the best consensus $N_{sat} = 10$ and $N_{dis} = 0$. To select a small subset of solutions, we use the "min" operator for conjunction as in Equation 20. The first ranked consensus solutions are provided in Table 7.

The values $\sigma_{Pi}(ZG_{ki}, B_i)$ from Equation 19 are shown in Table A8 (see Appendix B). These represent the level of satisfaction for each DM.

**Remark 9.** *The example has illustrated the importance of the assumptions on which the proposal is based, the way to build and use both model of multi-criteria preferences, the search for optimum consensus solutions, the handle of imperfect information through interval numbers, and the modeling of conservatism from individual DMs when they face this imperfect information. For simplicity and space consumption reasons, the example assumed that all the DMs face the same set of objective functions, and consider the same resource consumption per project and the same resource availability, although the DMs may have their own conservatism attitude concerning these issues; the ill-definition of proper collective judgments is modeled by interval numbers.*

**Table 6.** Best consensus solution to Problem 1 with $N_{sat} = 6$ and $N_{dis} = 0$.

| O1 | | O2 | | O3 | | O4 | | O5 | | O6 | | O7 | | O8 | | O9 | |
|---|---|---|---|---|---|---|---|---|---|---|---|---|---|---|---|---|---|
| 1220175 | 1321785 | 1014920 | 1099490 | 1376910 | 1491640 | 893130 | 967550 | 1660130 | 1798520 | 1172815 | 1270545 | 1897530 | 2055690 | 1450070 | 1649190 | 1470300 | 1592790 |

**Table 7.** Best ranked solutions to Problem 1 with $N_{sat} = 10$ and $N_{dis} = 0$ (after the consensus round).

| O1 | | O2 | | O3 | | O4 | | O5 | | O6 | | O7 | | O8 | | O9 | | $\mu_{sat}$ |
|---|---|---|---|---|---|---|---|---|---|---|---|---|---|---|---|---|---|---|
| 1,223,260 | 1,325,180 | 1,026,360 | 1,111,900 | 1,455,145 | 1,576,405 | 905,975 | 981,465 | 1,590,645 | 1,723,195 | 1,201,815 | 1,301,915 | 1,980,240 | 2,145,280 | 1,458,810 | 1,580,330 | 1,603,185 | 1,736,745 | 0.7087 |
| 1,205,000 | 1,305,400 | 1,023,920 | 1,109,250 | 1,452,195 | 1,573,215 | 918,285 | 994,805 | 1,574,665 | 1,705,875 | 1,218,895 | 1,320,415 | 1,963,310 | 2,126,930 | 1,451,450 | 1,572,360 | 1,605,365 | 1,739,105 | 0.7069 |
| 1,218,835 | 1,320,385 | 1,000,750 | 1,084,130 | 1,486,750 | 1,610,620 | 902,530 | 977,760 | 1,577,835 | 1,709,335 | 1,223,220 | 1,325,120 | 1,886,760 | 2,044,020 | 1,436,160 | 1,555,800 | 1,630,050 | 1,765,860 | 0.7026 |
| 1,209,115 | 1,309,855 | 1,006,320 | 1,090,170 | 1,500,955 | 1,626,005 | 906,960 | 982,560 | 1,558,585 | 1,688,475 | 1,253,740 | 1,358,180 | 1,901,730 | 2,060,230 | 1,428,970 | 1,548,010 | 1,597,895 | 1,731,025 | 0.7019 |
| 1,218,590 | 1,320,110 | 1,020,780 | 1,105,840 | 1,434,715 | 1,554,285 | 929,865 | 1,007,325 | 1,618,785 | 1,753,695 | 1,226,165 | 1,328,295 | 1,895,680 | 2,053,670 | 1,468,330 | 1,590,660 | 1,582,605 | 1,714,455 | 0.7015 |

## 8. Concluding Remarks

The criticisms to GDM-MOP approaches discussed in the introduction have been basically overcome. To our knowledge, this paper has presented one of the most comprehensive approaches to group multi-objective optimization under imperfect information. Its remarkable generality is supported by several important features: (a) some group members may be compatible with an outranking model of preferences, but other ones may feel comfortable with a preference model based on a weighted-sum value function; (b) the use of outranking models allows to handle veto situations, incomparability and intransitive preferences, whereas the value function model is more appropriate for compensatory aggregation of preferences; (c) some (even all) objective values maybe not the same for different group members; (d) group members may consider their own set of objective functions and constraints; (e) objective values may be imprecise or uncertain; (f) imperfect information on available resources and requirements may be handled; and (g) each group member may have their own perception about the availability of resources and the requirement of resources per activity. It should be remarked that, although the problem was formulated in terms of maximization with resource constraints as in project portfolio optimization, the extension of the method to other formulations is straightforward.

Another important feature is related to the definition of what a satisfied group member is. Unlike other group decision making approaches, that definition is independent of the satisfaction stage of other group members. In this paper, the group members' stage of satisfaction/dissatisfaction is assigned by using multi-criteria ordinal classification approaches. These assignments fulfill several consistency properties whose theoretical proofs have been provided by the paper. The level of group satisfaction is represented by a bi-objective measure containing the number of group members who are satisfied/dissatisfied with each particular solution. Taking as bases (i) a limiting boundary (set by each group member) between classes "satisfactory" and "unsatisfactory", (ii) an outranking/preference relation, and (iii) an algorithm able to optimize the bi-objective measure of satisfaction/dissatisfaction, the method is capable of finding (within the decision variable space) the highest group satisfaction related to each particular stage of the consensus reaching process.

The whole approach is based on seven reasonable assumptions. The first two assumptions are common to both models of preferences (outranking and weighted-sum function). Assumptions 3–5 support the outranking model, whereas Assumptions 6–7 are basic for the functional model. Assumptions 4 and 7 show only a very slight difference; Assumptions 3 and 6 play the same role in each model of multi-criteria preferences.

In our approach, a good consensus is associated with a high level of satisfaction and a very low level of dissatisfaction. If the bi-objective measure of collective satisfaction/dissatisfaction is not judged as sufficiently close to the desired consensus level, then the group should perform an additional consensus round in which preference parameters, judgments, beliefs, and attitudes facing imperfect knowledge, should be closed. If, after several consensus rounds, the bi-objective measure of collective satisfaction/dissatisfaction does not reach the desired consensus level, the group (or the group moderator) should make a final decision among the different non-dominated solutions in the space of satisfaction/dissatisfaction.

To reach high consensus levels requires collaborative attitudes from the group members; this is frequent in project portfolio optimization since the individual DMs represent the common organizational interest. However, as a consequence of the independence of judgments about satisfaction/dissatisfaction, the method can work even in cases of very conflicting views from the individual DMs, helping to identify the best possible partial consensus. During discussions and exchange of opinions, the group members should report their real preferences and beliefs. There is no way to manipulate the consensus solution maliciously by declaring insincere preferences and beliefs.

The present proposal is capable of modeling priorities on the set of group members; when the moderator has sufficient authority, (s)he can weigh the different non-dominated

solutions (group satisfaction, group dissatisfaction) according to who are the satisfied and dissatisfied group members.

The method (with both models of preferences) behaved well in an illustrative group many-objective project portfolio optimization problem. For simplicity, we used the same parameters for both models, what is a rough approximation since the weights have different meanings. The ideal consensus was achieved with a single consensus round. There is no obstacle to the combined use of both preference models.

Another interesting feature of our approach is its potential to handle large scale GDM-MOPs. If parallel processing is used in solving Problem 11, the computational cost of the proposal depends weakly on the number of DMs (see Remark 7.ii in Section 6). With only minor changes, the method could be adapted to handle such optimization problems as an avenue of future research.

Other directions for future research are the following:

- Since consensus is affected by intense (dis)satisfaction, we require to model also high satisfaction and strong dissatisfaction. This could be addressed by multi-criteria ordinal classification methods, but the model would be more complex due to the increment of classes.
- Development of models for making an appropriate selection of the best consensus among non-dominated solutions of Problem 1. Logic-based models representing predicates like "a strong majority is satisfied with ..." and "a very weak minority disagrees with ..." may be used to select one of the non-dominated solutions in the space of collective satisfaction/dissatisfaction. This would permit to reduce, perhaps replace, the role of the moderator in choosing the final decision. These models should be able to reflect intense satisfaction and dissatisfaction from the group members.
- To alleviate the DM hard cognitive task in assessing limiting boundaries, this is more relevant as the numbers of objective functions and DMs increase.

**Author Contributions:** Conceptualization, E.F., L.C.-R. and N.R.-V.; methodology, E.F., C.G.-S, N.R.-V. and L.C.-R.; software, N.R.-V. and C.G.-S.; validation, E.F., L.C.-R., N.R.-V. and C.G.-S.; formal analysis, E.F., L.C.-R., C.G.-S.; investigation, L.C.-R., E.F., N.R.-V. and C.G.-S.; writing—original draft preparation, E.F., C.G.-S. and N.R.-V.; writing—review and editing, C.G.-S., L.C.-R., E.F. and N.R.-V. All authors have read and agreed to the published version of the manuscript.

**Funding:** This research received no external funding.

**Institutional Review Board Statement:** Not applicable.

**Informed Consent Statement:** Not applicable.

**Data Availability Statement:** The instances and other files used here are available at https://www.dropbox.com/s/t80u9kbdcub6jua/Projects%20Description.pdf?dl=0 (accessed on 12 May 2021).

**Acknowledgments:** Authors thanks to CONACYT for supporting the projects from (a) Cátedras CONACYT Program with Number 3058. (b) CONACYT Project with Number A1-S-11012 from Convocatoria de Investigación Científica Básica 2017–2018 and CONACYT Project with Number 312397 from Programa de Apoyo para Actividades Científicas, Tecnológicas y de Innovación (PAACTI), (c) CONACYT Project with Number 236154 and (d) CONACYT Project with Number 280081.

**Conflicts of Interest:** The authors declare no conflict of interest.

## Appendix A. The Extended I-NOSGA Method

The I-NOSGA algorithm was proposed by Balderas et al. (2019). Using an outranking relation, this method identifies non-strictly outranked portfolios (modeled using interval mathematics) during the evolutionary search, satisfying a set of constraints that can be described by intervals. In the present paper, we use a variant of the original algorithm.

The pseudo-code of the main loop of I-NOSGA is shown in Algorithm A1. The present work slightly modifies the algorithm to allow a generalization that admits the use of different models of preferences (*PM*), and strength measures (*SM*), which are appropriate for the specific characteristics of each decision maker; such elements of the algorithm are used to define the preference relation that supports the incorporation of the DM's preferences in the evolutionary search. The experiments in this paper explore two different *PM*s, one based on outranking, and the other based on a weighted-sum value function; each model is associated with its *SM*.

The Algorithm A1 starts by combining the existing population of parents and children in Line 1; then, using the specified preference model *PM*, the algorithm builds the non-outranked fronts (see Line 2). The fronts *F* formed in this way are used to create the new generation of parents $Pop_{T+1}$ (see Lines 5–11). The algorithm orderly includes complete fronts in $Pop_{T+1}$ (Line 6) and complete it with the best solutions from the last front $F_i$ that does not fit entirely; the solutions are taken in order according to the specified *SM* (see Lines 9 to 11). Finally, a new generation of individuals $QT+1$ is evolved from $Pop_{T+1}$ using the genetic operators chosen for this purpose (see Line 12).

---

**Algorithm A1. Interval Non-Outranked Sorting Genetic Algorithm**

**Input:**
$Pop_T$, the population of parents,
$Q_T$, the children generated in the previous iteration,
*PM*, the binary preference model used to compare pairs of solutions (*x,y*)
*SM*, the computation model for the strength measure of built solutions
**Output:** Next generation of parents $Pop_{T+1}$ and children $Q_{T+1}$

| | |
|---|---|
| 01: $R_T = Pop_T \cup Q_T$ | //combine parents and children population |
| 02: $F$ = sort-by-preferences ($R_T$, $\lvert Pop_T \rvert$, *PM*) | //create outrank **fronts** $F = (F_0, F_1, \dots)$ from $R_T$ using *PM* |
| 03: $Pop_{T+1} = \varnothing$ | //initialize new population $Pop_{T+1}$ |
| 04: $i = 0$ | |
| 05: **while** $\lvert Pop_{T+1} \rvert + \lvert F_i \rvert \leq N$ **do** | //fill the **new population** set $Pop_{T+1}$ |
| 06: $Pop_{T+1} = Pop_{T+1} \cup F_i$ | //**include front** $F_i$ that fits completely in $Pop_{T+1}$ |
| 07: $i = i + 1$ | //move to **next front** in the order set $F$ |
| 08: **end** | |
| 09: $FS$ = strength- assignment($F_i$, *PM*, *NSF*) | //measures the **strength** of the solutions in $F_i$, $NSF = F_0$ |
| 10: $F'_i$ = SORT($F_i$, *FS*) | //**sort** solutions in $F_i$ **by FS** in descending order |
| 11: $Pop_{T+1} = Pop_{T+1} \cup F'_i[1{:}N{-}\lvert Pop_{T+1} \rvert]$ | //**complete** $Pop_{T+1}$ with best solutions in $F'_I$ when $\lvert Pop_{T+1} \rvert < N$ |
| 12: $Q_{T+1}$ = make-new-pop($Pop_{T+1}$) | //construct **next generation** of children $Q_{T+1}$ using $Pop_{T+1}$ and the chosen **operators for selection, crossover** and **mutation** |
| 13: $T = T + 1$ | //iterate |

---

Observe that the pseudocode presented in Algorithm A1 reflects the main loop of I-NOSGA, i.e., the actions that are repeated until a user-defined stop criterion is met. In addition, note that the initialization strategies for the parent PopT and children QT populations are left to the user, and the best solutions will be in the front F0 of the last iteration.

Figure A1 represents the individual encoding (chromosome) used by the evolutionary algorithm. This is a binary array *x* of size *n* (the number of projects) in which the *i*-th element (*gen*) contains 1 if the *i*-th project belongs to the portfolio and 0 otherwise. Parents are chosen by binary tournament; the solution belonging to the better front is selected for crossover; ties are broken randomly. The implemented strategy used a one-point crossover and flip mutation. The mutation probability for each allele was set to 0.02.

| 1 | 0 | 1 | 1 | 1 | 0 | 0 | 0 | 1 | 1 |
|---|---|---|---|---|---|---|---|---|---|

**Figure A1.** Binary representation of a portfolio in the decision variable space.

Table A1 details the worst-case time complexity of the main loop of I-NOSGA. The analysis considers the variables *S*, *N*, and *P* as the size of the population, the number of criteria, and the number of candidate projects, respectively.

**Table A1.** I-NOSGA time complexity analysis.

|  | Complexity |
|---|---|
| 01: $R_T = Pop_T \cup Q_T$ | $O(S)$ |
| 02: $F$ = sort-by-preferences ($R_T$, $\lvert Pop_T \rvert$, $PM$) | $O(N^2 S^2)$ * |
| 03: $Pop_{T+1} = \varnothing$ | $O(1)$ |
| 04: $i = 0$ | $O(1)$ |
| 05: while $\lvert Pop_{T+1} \rvert + \lvert F_i \rvert \leq N$ do | $O(S)$ |
| 06: $Pop_{T+1} = Pop_{T+1} \cup F_i$ | $O(S)$ |
| 07: $i = i + 1$ | $O(1)$ |
| 08: end |  |
| 09: $FS$ = strength-assignment($F_i$, $PM$, $NSF$) | $O(S)$ |
| 10: $F'i$ = SORT($F_i$, $FS$) | $O(S^2)$ |
| 11: $Pop_{T+1} = Pop_{T+1} \cup F'i[1:N\text{-}\lvert Pop_{T+1} \rvert]$ | $O(S)$ |
| 12: $Q_{T+1}$ = make-new-pop($Pop_{T+1}$) | $O(NPS)$ |
| 13: $T = T + 1$ | $O(1)$ |

* Using the fast non-dominated sort proposed by Deb.

Let's point out that, based on the analysis shown in Table A1, the complexity of I-NOSGA linearly scales w.r.t. to the number of candidate projects $P$, i.e., its complexity is $O(P)$. This complexity plays a vital role because there might be thousands of candidate projects in some realistic scenarios.

## Appendix B. Updated Budget Requirements

**Table A2.** Parameters values of the model: (a) weight, (b) veto thresholds and (c) credibility and majority thresholds.

| (a) | | | | | | | | | | | | | | | | | | |
|---|---|---|---|---|---|---|---|---|---|---|---|---|---|---|---|---|---|---|
| DM | O1 | | O2 | | O3 | | O4 | | O5 | | O6 | | O7 | | O8 | | O9 | |
| 1 | 0.066 | 0.142 | 0.108 | 0.16 | 0.064 | 0.134 | 0.096 | 0.162 | 0.05 | 0.083 | 0.092 | 0.162 | 0.078 | 0.128 | 0.053 | 0.081 | 0.138 | 0.211 |
| 2 | 0.065 | 0.14 | 0.109 | 0.163 | 0.078 | 0.135 | 0.095 | 0.165 | 0.054 | 0.08 | 0.106 | 0.169 | 0.066 | 0.143 | 0.054 | 0.082 | 0.145 | 0.25 |
| 3 | 0.064 | 0.126 | 0.104 | 0.178 | 0.064 | 0.13 | 0.084 | 0.14 | 0.053 | 0.082 | 0.089 | 0.17 | 0.078 | 0.12 | 0.055 | 0.085 | 0.15 | 0.231 |
| 4 | 0.064 | 0.141 | 0.096 | 0.163 | 0.065 | 0.128 | 0.094 | 0.168 | 0.053 | 0.092 | 0.092 | 0.175 | 0.08 | 0.138 | 0.055 | 0.09 | 0.145 | 0.222 |
| 5 | 0.08 | 0.136 | 0.09 | 0.18 | 0.077 | 0.14 | 0.085 | 0.155 | 0.051 | 0.083 | 0.102 | 0.155 | 0.075 | 0.14 | 0.06 | 0.096 | 0.146 | 0.216 |
| 6 | 0.073 | 0.121 | 0.098 | 0.153 | 0.071 | 0.13 | 0.1 | 0.151 | 0.059 | 0.085 | 0.096 | 0.173 | 0.073 | 0.135 | 0.048 | 0.093 | 0.12 | 0.219 |
| 7 | 0.077 | 0.123 | 0.096 | 0.158 | 0.077 | 0.138 | 0.097 | 0.159 | 0.053 | 0.09 | 0.092 | 0.16 | 0.078 | 0.123 | 0.048 | 0.083 | 0.142 | 0.245 |
| 8 | 0.068 | 0.143 | 0.107 | 0.17 | 0.067 | 0.139 | 0.086 | 0.158 | 0.059 | 0.089 | 0.109 | 0.169 | 0.076 | 0.13 | 0.05 | 0.086 | 0.13 | 0.243 |
| 9 | 0.07 | 0.135 | 0.093 | 0.177 | 0.068 | 0.137 | 0.09 | 0.148 | 0.053 | 0.082 | 0.103 | 0.169 | 0.079 | 0.127 | 0.052 | 0.084 | 0.15 | 0.223 |
| 10 | 0.076 | 0.13 | 0.09 | 0.154 | 0.069 | 0.136 | 0.087 | 0.147 | 0.052 | 0.095 | 0.099 | 0.176 | 0.08 | 0.128 | 0.053 | 0.091 | 0.132 | 0.234 |

| (b) | | | | | | | | | | | | | | | | | | |
|---|---|---|---|---|---|---|---|---|---|---|---|---|---|---|---|---|---|---|
| DM | O1 | | O2 | | O3 | | O4 | | O5 | | O6 | | O7 | | O8 | | O9 | |
| 1 | 195,989 | 195,989 | 234,398 | 234,398 | 169,919 | 169,919 | 169,919 | 169,919 | 169,919 | 169,919 | 169,919 | 169,919 | 169,919 | 169,919 | 169,919 | 169,919 | 301,255 | 301,255 |
| 2 | 256,187 | 256,187 | 245,076 | 245,076 | 298,408 | 298,408 | 298,408 | 298,408 | 298,408 | 298,408 | 298,408 | 298,408 | 298,408 | 298,408 | 298,408 | 298,408 | 329,296 | 329,296 |
| 3 | 195,544 | 195,544 | 132,601 | 132,601 | 205,307 | 205,307 | 205,307 | 205,307 | 205,307 | 205,307 | 205,307 | 205,307 | 205,307 | 205,307 | 205,307 | 205,307 | 378,402 | 378,402 |
| 4 | 126,797 | 126,797 | 141,914 | 141,914 | 199,808 | 199,808 | 199,808 | 199,808 | 199,808 | 199,808 | 199,808 | 199,808 | 199,808 | 199,808 | 199,808 | 199,808 | 226,441 | 226,441 |
| 5 | 222,376 | 222,376 | 173,008 | 173,008 | 207,355 | 207,355 | 207,355 | 207,355 | 207,355 | 207,355 | 207,355 | 207,355 | 207,355 | 207,355 | 207,355 | 207,355 | 254,049 | 254,049 |
| 6 | 276,395 | 276,395 | 231,282 | 231,282 | 173,649 | 173,649 | 173,649 | 173,649 | 173,649 | 173,649 | 173,649 | 173,649 | 173,649 | 173,649 | 173,649 | 173,649 | 262,777 | 262,777 |
| 7 | 184,680 | 184,680 | 161,316 | 161,316 | 159,873 | 159,873 | 159,873 | 159,873 | 159,873 | 159,873 | 159,873 | 159,873 | 159,873 | 159,873 | 159,873 | 159,873 | 189,960 | 189,960 |
| 8 | 129,004 | 129,004 | 194,205 | 194,205 | 224,768 | 224,768 | 224,768 | 224,768 | 224,768 | 224,768 | 224,768 | 224,768 | 224,768 | 224,768 | 224,768 | 224,768 | 284,196 | 284,196 |
| 9 | 265,641 | 265,641 | 140,995 | 140,995 | 249,798 | 249,798 | 249,798 | 249,798 | 249,798 | 249,798 | 249,798 | 249,798 | 249,798 | 249,798 | 249,798 | 249,798 | 375,703 | 375,703 |
| 10 | 150,909 | 150,909 | 194,222 | 194,222 | 185,065 | 185,065 | 185,065 | 185,065 | 185,065 | 185,065 | 185,065 | 185,065 | 185,065 | 185,065 | 185,065 | 185,065 | 275,222 | 275,222 |

| (c) | | | | | |
|---|---|---|---|---|---|
| DM | $\alpha$ | $\xi$ | $\Lambda$ | | $B$ |
| 1 | 0.75 | 0.75 | 0.51 | 0.67 | 0.66 | 0.77 |
| 2 | 0.67 | 0.67 | 0.51 | 0.67 | 0.60 | 0.70 |
| 3 | 0.65 | 0.67 | 0.51 | 0.67 | 0.60 | 0.67 |
| 4 | 0.66 | 0.67 | 0.51 | 0.67 | 0.60 | 0.67 |
| 5 | 0.68 | 0.70 | 0.51 | 0.67 | 0.60 | 0.70 |
| 6 | 0.74 | 0.75 | 0.51 | 0.67 | 0.66 | 0.76 |
| 7 | 0.75 | 0.75 | 0.51 | 0.67 | 0.66 | 0.77 |
| 8 | 0.77 | 0.78 | 0.51 | 0.67 | 0.66 | 0.78 |
| 9 | 0.78 | 0.80 | 0.51 | 0.67 | 0.67 | 0.80 |
| 10 | 0.73 | 0.75 | 0.51 | 0.67 | 0.65 | 0.75 |

In Table A2c the notation is the following.

- $\alpha$: credibility threshold for dominance
- $\xi$: credibility threshold for sufficiency of resources
- $\lambda$: interval majority threshold
- $\beta$: credibility threshold for the crisp interval outranking.

**Table A3.** Limiting boundaries.

| | | | | | | Frontiers B | | | | |
|---|---|---|---|---|---|---|---|---|---|---|
| | | O1 | O2 | O3 | O4 | O5 | O6 | O7 | O8 | O9 |
| DM1 | $b_1$ | 1,234,925 | 1,036,475 | 1,417,065 | 968,685 | 1,570,565 | 1,211,330 | 1,925,345 | 1,365,315 | 1,658,795 |
| | $b_2$ | 1,251,926 | 1,036,475 | 1,417,065 | 968,685 | 1,570,565 | 1,211,330 | 1,891,403 | 1,365,315 | 1,658,795 |
| | $b_3$ | 1,234,925 | 1,036,475 | 1,342,175 | 968,685 | 1,570,565 | 1,234,373 | 1,925,345 | 1,365,315 | 1,658,795 |
| DM2 | $b_1$ | 1,259,905 | 975,365 | 1,474,745 | 999,450 | 1,542,150 | 1,279,330 | 1,983,810 | 1,486,095 | 1,633,700 |
| | $b_2$ | 1,296,052 | 975,365 | 1,474,745 | 999,450 | 1,542,150 | 1,230,224 | 1,983,810 | 1,486,095 | 1,633,700 |
| | $b_3$ | 1,259,905 | 975,365 | 1,475,379 | 999,450 | 1,542,150 | 1,279,330 | 1,983,810 | 1,374,761 | 1,633,700 |
| DM3 | $b_1$ | 1,291,140 | 1,024,090 | 1,509,600 | 924,815 | 1,712,030 | 1,255,190 | 2,009,685 | 1,358,450 | 1,661,035 |
| | $b_2$ | 1,291,140 | 1,024,090 | 1,509,600 | 924,815 | 1,712,030 | 1,278,324 | 2,009,685 | 1,276,835 | 1,661,035 |
| | $b_3$ | 1,291,140 | 9,725,56 | 1,509,600 | 924,815 | 1,712,030 | 1,255,190 | 2,106,229 | 1,358,450 | 1,661,035 |
| DM4 | $b_1$ | 1,189,995 | 1,060,215 | 1425755 | 977,190 | 1,667,295 | 1,312,095 | 2,067,265 | 1,370,165 | 1,610,035 |
| | $b_2$ | 1,189,995 | 982,445 | 1441611 | 977,190 | 1,667,295 | 1,312,095 | 2,067,265 | 1,370,165 | 1,610,035 |
| | $b_3$ | 1,212,503 | 1,060,215 | 1425755 | 977,190 | 1,667,295 | 1,266,805 | 2,067,265 | 1,370,165 | 1,610,035 |
| DM5 | $b_1$ | 1,215,945 | 1,010,400 | 1,488,700 | 927,465 | 1,702,140 | 1,234,430 | 1,992,975 | 1,383,445 | 1,715,605 |
| | $b_2$ | 1,230,005 | 1,010,400 | 1,488,700 | 927,465 | 1,702,140 | 1,234,430 | 1,936,667 | 1,383,445 | 1,715,605 |
| | $b_3$ | 1,215,945 | 1,033,867 | 1,488,700 | 927,465 | 1,702,140 | 1,234,430 | 1,992,975 | 1,268,806 | 1,715,605 |
| DM6 | $b_1$ | 1,169,800 | 1,057,620 | 1,502,780 | 962,570 | 1,652,975 | 1,211,310 | 2,012,250 | 1,445,540 | 1,576,135 |
| | $b_2$ | 1,169,800 | 1,057,620 | 1,502,780 | 962,570 | 1,652,975 | 1,211,310 | 1,821,482 | 1,448,256 | 1,576,135 |
| | $b_3$ | 1,170,982 | 1,057,620 | 1,502,780 | 962,570 | 1,652,975 | 1,211,310 | 2,012,250 | 1,445,540 | 1,420,488 |
| DM7 | $b_1$ | 1,211,085 | 1,026,425 | 1,394,220 | 910,105 | 1,588,335 | 1,266,775 | 2,029,310 | 1,414,960 | 1,663,015 |
| | $b_2$ | 1,153,127 | 1,026,425 | 1,394,220 | 921,731 | 1,588,335 | 1,266,775 | 2,029,310 | 1,414,960 | 1,663,015 |
| | $b_3$ | 1,211,085 | 1,037,823 | 1,394,220 | 910,105 | 1,576,124 | 1,266,775 | 2,029,310 | 1,414,960 | 1,663,015 |
| DM8 | $b_1$ | 1,297,695 | 1,009,480 | 1,471,245 | 946,720 | 1,538,740 | 1,231,455 | 1,820,550 | 1,489,320 | 1,596,360 |
| | $b_2$ | 1,297,695 | 1,009,480 | 1,358,643 | 946,720 | 1,538,740 | 1,231,455 | 1,820,550 | 1,489,320 | 1,598,868 |
| | $b_3$ | 1,297,695 | 1,009,480 | 1,471,245 | 933,477 | 1,541,083 | 1,231,455 | 1,820,550 | 1,489,320 | 1,596,360 |
| DM9 | $b_1$ | 1,139,045 | 1,077,665 | 1,377,385 | 925,280 | 1,570,825 | 1,215,730 | 1,982,670 | 1,429,490 | 1,584,005 |
| | $b_2$ | 1,139,045 | 1,077,665 | 1,377,385 | 925,280 | 1,570,825 | 1,154,986 | 1,982,670 | 1,429,490 | 1,606,499 |
| | $b_3$ | 1,142,960 | 1,077,665 | 1,377,385 | 925,280 | 1,570,825 | 1,215,730 | 1,982,670 | 1,358,901 | 1,584,005 |
| DM10 | $b_2$ | 1,262,100 | 1,030,870 | 1,442,840 | 931,350 | 1,629,665 | 1,251,900 | 1,932,615 | 1,450,675 | 1,628,245 |
| | $b_3$ | 1,262,100 | 1,030,870 | 1,442,840 | 931,350 | 1,629,665 | 1,251,900 | 1,959,614 | 1,450,675 | 1,540,127 |
| | $b_1$ | 1,262,100 | 975,097 | 1,442,840 | 953,011 | 1,629,665 | 1,251,900 | 1,932,615 | 1,450,675 | 1,628,245 |

**Table A4.** The updated (a) weights after the consensus round, (b) veto thresholds after the consensus round, and (c) credibility thresholds after the consensus round.

| | | | | | (a) | | | | | | | | | | | | | |
|---|---|---|---|---|---|---|---|---|---|---|---|---|---|---|---|---|---|---|
| DM | O1 | | O2 | | O3 | | O4 | | O5 | | O6 | | O7 | | O8 | | O9 | |
| 1 | 0.068 | 0.100 | 0.105 | 0.143 | 0.072 | 0.108 | 0.098 | 0.133 | 0.057 | 0.077 | 0.098 | 0.135 | 0.082 | 0.114 | 0.057 | 0.078 | 0.142 | 0.189 |
| 2 | 0.068 | 0.099 | 0.105 | 0.144 | 0.079 | 0.115 | 0.097 | 0.133 | 0.059 | 0.079 | 0.104 | 0.142 | 0.076 | 0.108 | 0.058 | 0.078 | 0.145 | 0.192 |
| 3 | 0.067 | 0.099 | 0.105 | 0.141 | 0.072 | 0.108 | 0.092 | 0.127 | 0.058 | 0.078 | 0.097 | 0.134 | 0.082 | 0.114 | 0.058 | 0.079 | 0.144 | 0.195 |
| 4 | 0.067 | 0.099 | 0.101 | 0.137 | 0.073 | 0.109 | 0.097 | 0.132 | 0.058 | 0.078 | 0.098 | 0.135 | 0.083 | 0.115 | 0.058 | 0.079 | 0.145 | 0.192 |
| 5 | 0.065 | 0.107 | 0.098 | 0.134 | 0.079 | 0.115 | 0.092 | 0.128 | 0.057 | 0.077 | 0.103 | 0.140 | 0.081 | 0.112 | 0.061 | 0.081 | 0.146 | 0.193 |
| 6 | 0.069 | 0.103 | 0.102 | 0.138 | 0.076 | 0.112 | 0.099 | 0.135 | 0.061 | 0.081 | 0.100 | 0.137 | 0.080 | 0.111 | 0.055 | 0.075 | 0.133 | 0.180 |
| 7 | 0.067 | 0.105 | 0.101 | 0.137 | 0.079 | 0.115 | 0.098 | 0.134 | 0.058 | 0.078 | 0.098 | 0.135 | 0.082 | 0.114 | 0.055 | 0.075 | 0.144 | 0.191 |
| 8 | 0.069 | 0.101 | 0.106 | 0.143 | 0.074 | 0.110 | 0.093 | 0.128 | 0.061 | 0.081 | 0.102 | 0.144 | 0.081 | 0.113 | 0.056 | 0.076 | 0.138 | 0.185 |
| 9 | 0.070 | 0.102 | 0.100 | 0.136 | 0.074 | 0.110 | 0.095 | 0.130 | 0.058 | 0.078 | 0.104 | 0.141 | 0.083 | 0.114 | 0.057 | 0.077 | 0.144 | 0.195 |
| 10 | 0.067 | 0.105 | 0.098 | 0.134 | 0.075 | 0.111 | 0.093 | 0.129 | 0.058 | 0.078 | 0.102 | 0.139 | 0.083 | 0.115 | 0.057 | 0.078 | 0.139 | 0.186 |

| | | | | | (b) | | | | | | | | | | | | | |
|---|---|---|---|---|---|---|---|---|---|---|---|---|---|---|---|---|---|---|
| DM | O1 | | O2 | | O3 | | O4 | | O5 | | O6 | | O7 | | O8 | | O9 | |
| 1 | 198,171 | 198,171 | 190,206 | 190,206 | 198,904 | 198,904 | 200,051 | 200,051 | 200,166 | 200,166 | 200,178 | 200,178 | 200,179 | 200,179 | 200,179 | 200,179 | 306,014 | 306,014 |
| 2 | 172,435 | 172,435 | 184,867 | 184,867 | 192,629 | 192,629 | 196,072 | 196,072 | 196,416 | 196,416 | 196,451 | 196,451 | 196,454 | 196,454 | 196,454 | 196,454 | 301,513 | 301,513 |
| 3 | 197,948 | 197,948 | 168,769 | 168,769 | 216,598 | 216,598 | 217,745 | 217,745 | 217,860 | 217,860 | 217,872 | 217,872 | 217,873 | 217,873 | 217,873 | 217,873 | 276,960 | 276,960 |
| 4 | 163,575 | 163,575 | 173,425 | 173,425 | 213,848 | 213,848 | 214,996 | 214,996 | 215,111 | 215,111 | 215,122 | 215,122 | 215,123 | 215,123 | 215,123 | 215,123 | 268,607 | 268,607 |
| 5 | 189,340 | 189,340 | 188,972 | 188,972 | 217,622 | 217,622 | 218,769 | 218,769 | 218,884 | 218,884 | 218,896 | 218,896 | 218,897 | 218,897 | 218,897 | 218,897 | 282,411 | 282,411 |
| 6 | 162,331 | 162,331 | 191,764 | 191,764 | 200,769 | 200,769 | 201,916 | 201,916 | 202,031 | 202,031 | 202,043 | 202,043 | 202,044 | 202,044 | 202,044 | 202,044 | 286,775 | 286,775 |
| 7 | 192,516 | 192,516 | 183,126 | 183,126 | 193,881 | 193,881 | 195,028 | 195,028 | 195,143 | 195,143 | 195,155 | 195,155 | 195,156 | 195,156 | 195,156 | 195,156 | 250,367 | 250,367 |
| 8 | 164,678 | 164,678 | 199,571 | 199,571 | 226,328 | 226,328 | 227,476 | 227,476 | 227,591 | 227,591 | 227,602 | 227,602 | 227,603 | 227,603 | 227,603 | 227,603 | 297,485 | 297,485 |
| 9 | 167,708 | 167,708 | 172,966 | 172,966 | 216,934 | 216,934 | 220,377 | 220,377 | 220,721 | 220,721 | 220,756 | 220,756 | 220,759 | 220,759 | 220,759 | 220,759 | 278,309 | 278,309 |
| 10 | 175,631 | 175,631 | 199,579 | 199,579 | 206,477 | 206,477 | 207,624 | 207,624 | 207,739 | 207,739 | 207,751 | 207,751 | 207,752 | 207,752 | 207,752 | 207,752 | 292,998 | 292,998 |

| | | (c) | | |
|---|---|---|---|---|
| DM | A | $\xi$ | $\beta$ | |
| 1 | 0.70 | 0.72 | 0.62 | 0.70 |
| 2 | 0.69 | 0.70 | 0.62 | 0.67 |
| 3 | 0.68 | 0.70 | 0.62 | 0.67 |
| 4 | 0.69 | 0.70 | 0.62 | 0.67 |
| 5 | 0.70 | 0.71 | 0.62 | 0.67 |
| 6 | 0.71 | 0.72 | 0.62 | 0.70 |
| 7 | 0.70 | 0.72 | 0.62 | 0.70 |
| 8 | 0.69 | 0.70 | 0.62 | 0.70 |
| 9 | 0.69 | 0.69 | 0.62 | 0.70 |
| 10 | 0.71 | 0.72 | 0.63 | 0.69 |

**Table A5.** Updated budget requirements after the consensus round.

| Project | Cost | | Project | Cost | | Project | Cost | | Project | Cost | |
|---------|------|------|---------|------|--------|---------|------|--------|---------|------|--------|
| 1 | 9260 | 9640 | 26 | 9522 | 9908 | 51 | 8197 | 8533 | 76 | 5472 | 5698 |
| 2 | 6235 | 6485 | 27 | 9697 | 10,093 | 52 | 5962 | 6208 | 77 | 9657 | 10,053 |
| 3 | 5772 | 6008 | 28 | 9535 | 9925 | 53 | 7450 | 7750 | 78 | 7450 | 7750 |
| 4 | 7665 | 7975 | 29 | 8222 | 8558 | 54 | 7327 | 7623 | 79 | 5072 | 5278 |
| 5 | 9362 | 9748 | 30 | 9012 | 9378 | 55 | 6542 | 6808 | 80 | 8260 | 8600 |
| 6 | 7410 | 7710 | 31 | 5972 | 6218 | 56 | 9727 | 10,123 | 81 | 6592 | 6858 |
| 7 | 7675 | 7985 | 32 | 9065 | 9435 | 57 | 5490 | 5710 | 82 | 7422 | 7728 |
| 8 | 9512 | 9898 | 33 | 5370 | 5590 | 58 | 8780 | 9140 | 83 | 6962 | 7248 |
| 9 | 7360 | 7660 | 34 | 9085 | 9455 | 59 | 7855 | 8175 | 84 | 8495 | 8845 |
| 10 | 5602 | 5828 | 35 | 8085 | 8415 | 60 | 6360 | 6620 | 85 | 5790 | 6030 |
| 11 | 7647 | 7963 | 36 | 5380 | 5600 | 61 | 6217 | 6473 | 86 | 7855 | 8175 |
| 12 | 4990 | 5190 | 37 | 7677 | 7993 | 62 | 5880 | 6120 | 87 | 8345 | 8685 |
| 13 | 5747 | 5983 | 38 | 9372 | 9758 | 63 | 5612 | 5838 | 88 | 6002 | 6248 |
| 14 | 8590 | 8940 | 39 | 7470 | 7770 | 64 | 9565 | 9955 | 89 | 7740 | 8060 |
| 15 | 7930 | 8250 | 40 | 6922 | 7208 | 65 | 8657 | 9013 | 90 | 9707 | 10,103 |
| 16 | 8045 | 8375 | 41 | 9012 | 9378 | 66 | 7890 | 8210 | 91 | 6000 | 6240 |
| 17 | 8410 | 8750 | 42 | 9412 | 9798 | 67 | 6565 | 6835 | 92 | 7392 | 7698 |
| 18 | 5387 | 5603 | 43 | 5032 | 5238 | 68 | 9767 | 10,163 | 93 | 5592 | 5818 |
| 19 | 6340 | 6600 | 44 | 7982 | 8308 | 69 | 8165 | 8495 | 94 | 9605 | 9995 |
| 20 | 7850 | 8170 | 45 | 6052 | 6298 | 70 | 6065 | 6315 | 95 | 6572 | 6838 |
| 21 | 9360 | 9740 | 46 | 9087 | 9463 | 71 | 8320 | 8660 | 96 | 5012 | 5218 |
| 22 | 8195 | 8525 | 47 | 7850 | 8170 | 72 | 6380 | 6640 | 97 | 8830 | 9190 |
| 23 | 5910 | 6150 | 48 | 6787 | 7063 | 73 | 9207 | 9583 | 98 | 5685 | 5915 |
| 24 | 5787 | 6023 | 49 | 6217 | 6473 | 74 | 9797 | 10,193 | 99 | 5377 | 5593 |
| 25 | 5237 | 5453 | 50 | 7760 | 8080 | 75 | 6052 | 6298 | 100 | 5695 | 5925 |

**Table A6.** Values of $\sigma_i(Z_{Gki}, B_i, \lambda_i)$ for the solutions in Table 4.

| Sol. | DM1 | DM2 | DM3 | DM4 | DM5 | DM6 | DM7 | DM8 | DM9 | DM10 |
|------|--------|--------|--------|--------|--------|--------|--------|--------|--------|--------|
| 1 | 0.9816 | 0.8796 | 0.8958 | 0.8364 | 0.8345 | 0.8422 | 0.8814 | 1.0000 | 0.9921 | 0.9386 |
| 2 | 0.8643 | 0.9764 | 0.8479 | 0.8068 | 0.8626 | 0.8161 | 1.0000 | 1.0000 | 1.0000 | 0.8892 |
| 3 | 0.9030 | 0.8196 | 0.8526 | 0.9027 | 0.8418 | 0.8129 | 0.9427 | 0.8529 | 0.9630 | 0.7984 |
| 4 | 0.8808 | 0.9054 | 0.7975 | 0.8904 | 0.7910 | 0.9399 | 0.8485 | 1.0000 | 0.9977 | 1.0000 |
| 5 | 1.0000 | 0.9025 | 0.8559 | 0.7986 | 0.7880 | 0.8374 | 0.8435 | 1.0000 | 1.0000 | 0.9423 |

**Table A7.** Limiting boundaries (three limiting solution for each DM).

| O1 | | O2 | | O3 | | O4 | | O5 | | O6 | | O7 | | O8 | | O9 | |
|---|---|---|---|---|---|---|---|---|---|---|---|---|---|---|---|---|---|
| 1,172,035 | 1,196,440 | 935,955 | 955,448 | 1,293,390 | 1,320,333 | 921,260 | 940,450 | 1,703,365 | 1,738,853 | 1,059,975 | 1,082,055 | 1,831,405 | 1,869,578 | 1,366,425 | 1,394,890 | 1,420,215 | 1,449,790 |
| 1,172,035 | 1,196,440 | 936,369 | 955,870 | 1,293,390 | 1,320,333 | 921,260 | 940,450 | 1,703,365 | 1,738,853 | 1,059,975 | 1,082,055 | 1,831,405 | 1,869,578 | 1,366,425 | 1,394,890 | 1,341,051 | 1,368,978 |
| 1,172,035 | 1,196,440 | 935,955 | 955,448 | 1,293,390 | 1,320,333 | 840,542 | 858,050 | 1,703,365 | 1,738,853 | 1,064,999 | 1,087,184 | 1,831,405 | 1,869,578 | 1,366,425 | 1,394,890 | 1,420,215 | 1,449,790 |
| 1,210,295 | 1,235,513 | 989,555 | 1,010,173 | 1,322,445 | 1,349,990 | 899,420 | 918,158 | 1,607,385 | 1,640,873 | 1,165,840 | 1,190,123 | 1,915,940 | 1,955,865 | 1,451,480 | 1,481,713 | 1,478,550 | 1,509,343 |
| 1,210,295 | 1,235,513 | 989,555 | 1,010,173 | 1,322,445 | 1,349,990 | 820,944 | 838,047 | 1,607,385 | 1,640,873 | 1,165,840 | 1,190,123 | 1,915,940 | 1,955,865 | 1,451,480 | 1,481,713 | 1,484,672 | 1,515,592 |
| 1,210,295 | 1,235,513 | 1,001,082 | 1,021,940 | 1,322,445 | 1,349,990 | 899,420 | 918,158 | 1,607,385 | 1,640,873 | 1,090,067 | 1,112,771 | 1,915,940 | 1,955,865 | 1,451,480 | 1,481,713 | 1,478,550 | 1,509,343 |
| 1,169,955 | 1,194,315 | 1,012,675 | 1,033,773 | 1,366,235 | 1,394,695 | 871,970 | 890,138 | 1,771,940 | 1,808,860 | 1,191,690 | 1,216,513 | 1,874,150 | 1,913,203 | 1,469,130 | 1,499,725 | 1,433,435 | 1,463,290 |
| 1,169,955 | 1,194,315 | 1,012,675 | 1,033,773 | 1,366,235 | 1,394,695 | 871,970 | 890,138 | 1,771,940 | 1,808,860 | 1,201,862 | 1,226,897 | 1,874,150 | 1,913,203 | 1,469,130 | 1,499,725 | 1,424,088 | 1,453,749 |
| 1,169,955 | 1,194,315 | 978,386 | 998,769 | 1,366,235 | 1,394,695 | 882,140 | 900,519 | 1,771,940 | 1,808,860 | 1,191,690 | 1,216,513 | 1,874,150 | 1,913,203 | 1,469,130 | 1,499,725 | 1,433,435 | 1,463,290 |
| 1,202,270 | 1,227,310 | 1,006,320 | 1,027,280 | 1,393,975 | 1,423,008 | 897,320 | 916,015 | 1,566,755 | 1,618,775 | 1,163,860 | 1,188,108 | 1,914,190 | 1,954,078 | 1,462,680 | 1,493,143 | 1,479,985 | 1,510,815 |
| 1,202,270 | 1,227,310 | 973,732 | 994,013 | 1,393,975 | 1,423,008 | 897,320 | 916,015 | 1,566,755 | 1,618,775 | 1,175,213 | 1,199,697 | 1,914,190 | 1,954,078 | 1,462,680 | 1,493,143 | 1,479,985 | 1,510,815 |
| 1,202,270 | 1,227,310 | 1,006,320 | 1,027,280 | 1,393,975 | 1,423,008 | 863,664 | 881,658 | 1,566,755 | 1,618,775 | 1,163,860 | 1,188,108 | 1,914,190 | 1,954,078 | 1,462,680 | 1,493,143 | 1,493,835 | 1,524,954 |
| 1,210,395 | 1,235,598 | 1,005,315 | 1,026,260 | 1,379,125 | 1,407,855 | 843,045 | 860,610 | 1,668,500 | 1,703,265 | 1,184,000 | 1,208,658 | 1,869,795 | 1,908,760 | 1,465,640 | 1,496,160 | 1,392,025 | 1,421,003 |
| 1,210,395 | 1,235,598 | 1,013,090 | 1,034,197 | 1,379,125 | 1,407,855 | 827,099 | 844,332 | 1,668,500 | 1,703,265 | 1,184,000 | 1,208,658 | 1,869,795 | 1,908,760 | 1,465,640 | 1,496,160 | 1,392,025 | 1,421,003 |
| 1,210,395 | 1,235,598 | 1,005,315 | 1,026,260 | 1,379,125 | 1,407,855 | 843,045 | 860,610 | 1,668,500 | 1,703,265 | 1,120,592 | 1,143,929 | 1,869,795 | 1,908,760 | 1,465,640 | 1,496,160 | 1,394,685 | 1,423,718 |
| 1,134,520 | 1,158,138 | 1,002,865 | 1,023,758 | 1,283,820 | 1,310,563 | 864,205 | 882,210 | 1,667,285 | 1,702,025 | 1,098,580 | 1,121,465 | 1,872,725 | 1,911,748 | 1,356,185 | 1,384,428 | 1,448,755 | 1,478,918 |
| 1,134,520 | 1,158,138 | 1,002,865 | 1,023,758 | 1,283,820 | 1,310,563 | 828,005 | 845,256 | 1,667,285 | 1,702,025 | 1,102,103 | 1,125,062 | 1,872,725 | 1,911,748 | 1,356,185 | 1,384,428 | 1,448,755 | 1,478,918 |
| 1,134,520 | 1,158,138 | 958,718 | 978,691 | 1,283,820 | 1,310,563 | 864,205 | 882,210 | 1,667,285 | 1,702,025 | 1,098,580 | 1,121,465 | 1,872,725 | 1,911,748 | 1,356,185 | 1,384,428 | 1,454,714 | 1,485,000 |
| 1,172,035 | 1,196,440 | 935,955 | 955,448 | 1,293,390 | 1,320,333 | 921,260 | 940,450 | 1,703,365 | 1,738,853 | 1,059,975 | 1,082,055 | 1,831,405 | 1,869,578 | 1,366,425 | 1,394,890 | 1,420,215 | 1,449,790 |
| 1,172,035 | 1,196,440 | 935,955 | 955,448 | 1,293,390 | 1,320,333 | 910,799 | 929,771 | 1,703,365 | 1,738,853 | 1,059,975 | 1,082,055 | 1,831,405 | 1,869,578 | 1,366,425 | 1,394,890 | 1,424,702 | 1,454,371 |
| 1,172,035 | 1,196,440 | 939,929 | 959,504 | 1,293,390 | 1,320,333 | 921,260 | 940,450 | 1,703,365 | 1,738,853 | 1,027,471 | 1,048,874 | 1,831,405 | 1,869,578 | 1,366,425 | 1,394,890 | 1,420,215 | 1,449,790 |
| 1,142,135 | 1,165,920 | 930,485 | 949,865 | 1,331435 | 1,359,165 | 876,760 | 895,023 | 1,619,515 | 1,671,175 | 1,147,440 | 1,171,338 | 1,905,355 | 1,945,068 | 1,465,220 | 1,495,733 | 1,451,000 | 1,481,223 |
| 1,142,135 | 1,165,920 | 931,023 | 950,415 | 1,331435 | 1,359,165 | 800,414 | 817,086 | 1,619,515 | 1,671,175 | 1,147,440 | 1,171,338 | 1,905,355 | 1,945,068 | 1,465,220 | 1,495,733 | 1,451,000 | 1,481,223 |
| 1,142,135 | 1,165,920 | 930,485 | 949,865 | 1,331435 | 1,359,165 | 876,760 | 895,023 | 1,619,515 | 1,671,175 | 1,111,037 | 1,134,176 | 1,905,355 | 1,945,068 | 1,465,220 | 1,495,733 | 1,453,084 | 1,483,350 |
| 1,157,660 | 1,181,773 | 981,860 | 1,002,313 | 1,363505 | 1,391,908 | 890,910 | 909,468 | 1,535,575 | 1,567,575 | 1,182,040 | 1,206,658 | 1,775,935 | 1,812,940 | 1,396,465 | 1,425,550 | 1,434,775 | 1,464,658 |
| 1,157,660 | 1,181,773 | 982,925 | 1,003,400 | 1,363505 | 1,391,908 | 809,103 | 825,956 | 1,535,575 | 1,567,575 | 1,182,040 | 1,206,658 | 1,775,935 | 1,812,940 | 1,396,465 | 1,425,550 | 1,434,775 | 1,464,658 |
| 1,157,660 | 1,181,773 | 981,860 | 1,002,313 | 1,363505 | 1,391,908 | 890,910 | 909,468 | 1,535,575 | 1,567,575 | 1,107,575 | 1,130,642 | 1,775,935 | 1,812,940 | 1,396,465 | 1,425,550 | 1,437,577 | 1,467,543 |
| 1,164,705 | 1,188,958 | 962,750 | 982,798 | 1,275670 | 1,302,250 | 860,665 | 878,600 | 1,605,355 | 1,638,810 | 1,141,455 | 1,165,230 | 1,822,560 | 1,860,530 | 1,452,580 | 1,482,835 | 1,411,810 | 1,441,215 |
| 1,164,705 | 1,188,958 | 876,601 | 894,854 | 1,275,670 | 1,302,250 | 860,665 | 878,600 | 1,605,355 | 1,638,810 | 1,141,455 | 1,165,230 | 1,822,560 | 1,860,530 | 1,452,580 | 1,482,835 | 1,419,115 | 1,448,672 |
| 1,164,705 | 1,188,958 | 962,750 | 982,798 | 1,275,670 | 1,302,250 | 826,801 | 844,030 | 1,605,355 | 1,638,810 | 1,142,690 | 1,166,491 | 1,822,560 | 1,860,530 | 1,452,580 | 1,482,835 | 1,411,810 | 1,441,215 |

**Table A8.** Values of $\sigma_{Pi}(Z_{Gki}, B_i)$ for the solutions in Table 7.

| Sol. | DM1 | DM2 | DM3 | DM4 | DM5 | DM6 | DM7 | DM8 | DM9 | DM10 |
|---|---|---|---|---|---|---|---|---|---|---|
| 1 | 0.7583 | 0.7467 | 0.7087 | 0.7391 | 0.7207 | 0.7674 | 0.7454 | 0.7805 | 0.7758 | 0.7963 |
| 2 | 0.7566 | 0.7453 | 0.7069 | 0.7374 | 0.7191 | 0.7658 | 0.7437 | 0.7789 | 0.7742 | 0.7947 |
| 3 | 0.7521 | 0.7417 | 0.7026 | 0.7330 | 0.7155 | 0.7615 | 0.7398 | 0.7746 | 0.7701 | 0.7905 |
| 4 | 0.7516 | 0.7413 | 0.7019 | 0.7324 | 0.7150 | 0.7613 | 0.7393 | 0.7743 | 0.7696 | 0.7902 |
| 5 | 0.7514 | 0.7404 | 0.7015 | 0.7320 | 0.7139 | 0.7610 | 0.7383 | 0.7740 | 0.7689 | 0.7895 |

### Appendix C. Description of the Variant of MOEA/D Used in Solving Problem 1

Zhang and Li in [59] proposed the decomposition algorithm MOEA/D to solve multi-objective optimization problems. The strategy decomposes a multi-objective problem into several scalar functions, which are subject to simultaneous optimization. Algorithm A2 depicts the pseudocode of a variant of MOEA/D, adapted to solving Problem 1 in the frame of project portfolio optimization. The methods—initialization and repair and improvement operator—make use of interval arithmetic to handle the ill-determination and imprecision that is present in parameters' values of preference models and constraints. The general idea behind the approach followed in the experiment used in this manuscript is better detailed in the remaining paragraphs of Appendix C.

---

**Algorithm A2. Variant of MOEA/D**

---

**Input:**
*N:* number of scalar functions,
*M*: number of objectives,
*Vector*: uniformly distributed set of vectors
$T = N/10$: size of neighborhood of weight vectors.

**Output:**
*EP*: approximatio of Pareto frontier.

01: $(x, z, FV, B(i)) \leftarrow Initializacion()$
02: **For** $i = 1$ **to** $N$ **do**
03:     $(x_k, x_l) \leftarrow RandomSelection(B(i), T)$
04:     $y \leftarrow OnePointCrossover(x_k, x_l)$
05:     $y' \leftarrow FlipMutation(y)$
06:     $y'' \leftarrow RepairAndImprovementOperator(y')$
07:     $UpdateSetZ(z, M, y'')$     //$z$: **for each** $j = 1, \ldots, m$, **if** $z_j < f_j(y')$ then set $z_j = f_j(y')$.
08:     $UpdateNeighborhood(B(i), FV, y'')$     // **for each** $j \in B(i)$, **if** $g^{te}(y' \mid V_{j,z})$ set $x_j = y'$ and $FV_j = F(y')$
09:     $UpdateEP(EP, y'')$     //Remove from EP all the vectors dominated by $F(y')$, and add $F(y')$ to EP if no vectors in EP dominate $F(y')$
10: **Stopping Criteria**: if maxEvaluations is reach, Otherwise, go to step 2.

---

The algorithm has as parameter inputs *N*, *M*, *Vector* = $\{V_1, V_2, \ldots, V_N\}$, and *T*. These parameters represent, respectively, the number of scalar functions or sub-problems in which the MOP has been divided, the number of objectives, a uniformly distributed set of size *N* containing weight vectors with two elements each (the weights were $V_i = \left(\frac{i}{N}, \frac{N-i}{N}\right)$, for each $V_i \in Vector$), and the size of the neighborhood of weight vectors. In addition, MOEA/D gives as output an approximation of the Pareto frontier (*EP*) formed by non-dominated solutions found during the optimization process. For this purpose, the algorithm works in two phases.

The first phase is the initialization phase (indicated in Line 0 of the Algorithm A2). Here the algorithm initializes properly the data structures required for the construction of the final set of solutions. These structures are: (1) the set *EP*, which will become the Pareto frontier approximation, and which initially is empty; (2) the neighborhood sets *B(i)* of each vector $V_i$ that contain the *T* closes weight vectors to $V_i$ by Euclidean distance; (3) the initial set of solutions $X = \{x_1, x_2, \ldots, x_N\}$ where each solution $x_j$, $1 \leq j \leq$ *Number of group members*, corresponds to the *j*-th DM best compromise in the decision variable space, and the remaining ones are randomly generated; (4) the set of fitness values $FV = \{FV_1, FV_2, \ldots, FV_N\}$, where each $FV_i$ is composed by the *M* objective values of each solution $x_i$; and 5) the set $z = \{z_1, \ldots, z_m\}$ formed by values $z_j$ corresponding to the best objective value among all the solutions built during the initialization process.

The second phase is an update process based on evolution. In this process, for each solution in the population, two solutions of the population are randomly selected and

used to generate a new solution by applying genetic operators. The operators used are the following ones:

One-point crossover: the two randomly chosen parents combine its chromosomes by designating a point in the bit string that represents them. The operator swaps between the two parents the bits in their encoding to the right of that point. The results are a new offspring of two children.

Flip mutation: This mutation process generates a random value for each allele in the bit string encoding the solution; every time that the generated value lies under a given probability (e.g., the probability of 0.02 used in this work), the bit value in the encoding is inverted.

After the application of the genetic operators, the created solution is subject to a repair and improvement phase, which is performed by the repair and improvement operator (*RIO*). The *RIO* combines the processes that might repair and improve a portfolio; in detail, it turns an infeasible solution into a quasi-feasible solution that is feasible with respect to the used budget but may not be feasible with respect to in the satisfaction degree of the DMs, i.e., it might not warranty that $N_{sat} \geq N_{dis}$. The *RIO* sorts the projects in the portfolio by the fitness ratio *FR*, and from the lowest *FR* to the highest, it eliminates projects one by one until the budget becomes feasible. After that, using a threshold parameter (set to 0.50) it also eliminates a proportion of the remaining projects. Then, from the highest FR to the lowest, the improvement algorithm adds projects into the portfolio while keeping the budget feasible. The fitness ratio *FR* is a generalized measure of the fitness of a solution based on its objectives; first, for the *i*-th project, the procedure computes the ratio $fc_{ij} = \textit{fitness}/\textit{cost}$ for each objective j; then, it computes $f_{ci}$ which is the median among the m values $fc_{ij}$ previously computed. The value $f_{cij}$ becomes the value of *FR*.

The final step is to check the stopping condition; if it is satisfied, the algorithm finishes and reports the *EP* as its output; otherwise, the algorithm returns to the second step. The stop criterion is fulfilled when a maximum number of evaluations of the objective function vector (100,000) is reached. During each iteration, the algorithm conveniently updates *B*, *EP*, *z*, and in the end, it returns the non-dominated set found in *EP*.

Lines 1 to 8 in Algorithm A2 describe the main loop of MOEA/D, and Table A9 details its time complexity based on a worst-case scenario. The analysis considers the variables *S*, *N*, *P*, and $n_g$ as the size of the population, the number of criteria on the project portfolio optimization, the number of candidate projects, and the number of DMs, respectively. The construction of the set *EP* was left outside the main loop.

**Table A9.** MOEA/D time complexity analysis.

| | Complexity |
|---|:---:|
| 0. $(x,z,FV,B(i)) \leftarrow$ Initialization() | |
| 1. For $i = 1$ to $S$ do | $O(S)$ |
| 2. $(x_k, x_l) \leftarrow$ RandomSelection($B(i),T$) | $O(1)$ |
| 3. $y \leftarrow$ OnePointCrossover($x_k, x_l$) | $O(P)$ |
| 4. $y' \leftarrow$ FlipMutation($y$) | $O(1)$ |
| 5. $y'' \leftarrow$ RepairAndImprovementOperator ($y'$) | $O(P)$ |
| 6. UpdateSetZ($z,M,y''$) | $O(N^2 n_g)$ |
| 7. UpdateNeighborhood($B(i),FV,y''$) | $O(S)$ |
| 8. UpdateEP($EP,y''$) | |
| 9. Stopping criteria: If maxEvaluations is reach, otherwise, go to Step 1. | |

Note that, based on the analysis shown in Table A9, the complexity of MOEA/D linearly scales w.r.t. to the number of candidate projects *P* and the number of DMs $n_g$ in the decision group. The linear growth in the number of members of a decision group favors this approach in scenarios with large groups of decision-makers.

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
