# Peer review of "A New Approach to Group Multi-Objective Optimization under Imperfect Information and Its Application to Project Portfolio Optimization"

_applsci, doi:10.3390/app11104575_

Round 1
Reviewer 1 Report
The authors provides an interesting paper that is suitable for publication after they address the following issues.
- The use of he/she or his/her throughout the paper inhibits readability. Use gender neutral terms such as they, their, decision maker etc.
- Seven Assumptions – Describe your serve assumptions in tabular form. Use the “What”, “so what”, and “therefor method” to describe your assumptions, why they are important, and how you use them in you methodology.
- Authors do not adequately describe methodology limitations in their paper. One cause for concern is your computational time required for generating you solutions. How does this time change in problems that are more complex? How well does your methodology scale to larger decision problems with more stakeholders?
- Use of tables in case study. The authors use several large tables in demonstrating their methodology. However, these tables provide limited information to the reader and increase the length of the article. Tables need to be revised and consolidated to improve understanding and readability.
Reviewer 2 Report
The subject is interesting. The introduction is discussed enough. Moreover, a suitable method is chosen and the content seems mathematically correct.
Although, there are lots of grammatical and misspelling problems. I strongly recommend checking the manuscript with a native mathematician.
Therefore, It can be recommended for publication after literature modification.
Reviewer 3 Report
Dear Authors!
Thank You for the relevant and interesting material.
The article is well structured and logically proved.
Please kindly shorten the introduction and clearly describe the research gap, research trask and the purpose of the article.
The literature review please kindly gether into a separate subsection 2.1.
In the subsection 2.2. please consider some theoretical aspects. The authors could be recommended adding some provisions about Thomas Saaty AHP and ANP. The authors should distinguish the suggested method from the famous methods including the mentioned Saaty and almost similar Multiplicative Lootsma method.
Some interesting issues arise regarding the optimization in modern conditions.
Probably some aspects could help the authors to develop new approach in the digital era:
1) J. Open Innov. Technol. Mark. Complex. 2021, 7(1), 59; https://doi.org/10.3390/joitmc7010059
2) Barykin, S.Y., Kapustina, I.V., Sergeev, S.M., Kalinina, O.V., Vilken, V.V., Poza, E.deL., ... Volkova, L.V. (2021). Developing the physical distribution digital twin model within the trade network. Academy of Strategic Management Journal, 20(S1).
Please form the Result section including all suggested models and calculation examples.
Discussion section should consists of some issues for further research taking into account also optimization problem in digital economy.
Round 2
Reviewer 1 Report
The authors' have made satisfactory revisions. Recommend publication.
Author Response
"Please see the attachment."
